# Comparative genomics-based development of a LAMP assay for rapid and reliable in-field detection of *Fusarium oxysporum* f. sp. cubense tropical race 4

**Mikel Arrieta Salgado[1,2]\***, **Diane Mostert[3]**, **Sebastien Ravel[1,2]**, **Samuel Rozsasi[3,4]**, **Veronique Maillot-Lebon[4]**, **Agus Sutanto****[5]**, **Catur Hermanto[5]**, **Mouzdalifa Mmadi[6]**, **Abdou Azali Hamza[6]**, **Nadia Adjanoh-Lubin[1,2]**, **Babitha Fenelon**[4], **Henri Adreit[1,2]**, **Sandrine Fabre[1,2]**, **Beatrix Coetzee[3]**, **J. Jansen van Vuuren**[3], **Sheryl Le Roux[3]**, **Altus Viljoen[3]**, **Yolande Chilin-Charles[2,7]**, **Jean Carlier[1,2]**, **Camilo Gianinazzi[8]**, **Emmanuel Wicker**[1,2]\*, **Isabelle Robène[3,4]\***

**1** CIRAD, UMR PHIM, Montpellier, France, **2** PHIM, Plant Health Institute, Univ Montpellier, INRAe, CIRAD, Institut Agro, Montpellier, France, **3** Department of Plant Pathology, Stellenbosch University, Stellenbosch, South Africa, **4** CIRAD, UMR PVBMT, Saint-Pierre, Réunion Island, France, **5** BRIN, Cibinong, Kabupaten Bogor, Indonesia, **6** Institut National de Recherche pour l'Agriculture, la Pêche et l'Environnement (INRAPE), Moroni, Union des Comores, **7** CIRAD, UMR PHIM, Capesterre-Belle-Eau, Guadeloupe, France, **8** QUALIPLANTE SAS, Laverune, France

\* mikarriet@gmail.com (MAS), emmanuel.wicker@cirad.fr (EW), isabelle.robene@cirad.fr (IR)

## Abstract

Fusarium wilt, caused by the soilborne fungus *Fusarium oxysporum* f. sp. *cubense* (Foc) Tropical race 4 (TR4) is currently devastating Cavendish banana production worldwide. The availability of accurate and rapid field-deployable detection tools is vital for effective disease prevention and containment practices, as no effective treatment is currently available. Design of reliable molecular detection tools, however, is challenged by genetic similarity between non-pathogenic and target Foc strains that are present in symptomatic plant material. The high genetic diversity present among Foc isolates further complicates the identification of unique genomic regions suitable for diagnosis. Current molecular assays are also still mostly confined to laboratory settings and rely on expensive equipment that needs skilled operators to perform. In the present study, a simple loop-mediated isothermal amplification (LAMP) detection assay was developed for direct in-field detection. A comprehensive database of Foc genomes including 148 TR4 genomes and 146 non-target genomes was used, considering the wide diversity within Foc. These genomic sequences were exploited in a comparative genomics *in silico* pipeline using the KEC method (K-mer elimination by cross-reference), combined with filtering steps using BLASTn, and Illumina read alignments. The specificity of the LAMP primers was validated using a diverse collection of 161 isolates, which included Foc-TR4 isolates, Foc VCGs, *F. oxysporum* spp. endophytes, other *Fusarium* spp., and genera which are commonly isolated from banana vascular tissue. The limit of detection reached $10^2$ copies µL$^{-1}$ of plasmid

**Data availability statement:** All relevant data are within the manuscript and its Supporting Information files.

**Funding:** This study was funded by the ANR-funded Projet France-Relance (PFR) CIRADQUALIPLANTE (2022-2024), the European Union Horizon 2020 Research and Innovation Program Marie Sklodowska-Curie Fellowship, through the INDICANTS project grant nr 890856, and co-funded by the European Union, the French Government and the « Région Réunion: L' Europe s'engage à La Réunion avec le FEADER », project grant REU77071-1-000011.

**Competing interests:** The authors have declared that non competing interests exist.

DNA, and the target pathogen was successfully identified both from artificially inoculated and naturally infested material in contrasted environments, indicating the LAMP assay to be a robust and reliable tool fully suitable for field diagnosis.

## Introduction

Fusarium wilt of banana is a soilborne vascular disease that threatens global banana production, and is considered one of the most destructive plant diseases worldwide [1–3]. The causal agent is *Fusarium oxysporum* f. sp. *cubense* (Foc), a highly diverse fungal species whose different strains have caused two major epidemics in the last century. Foc-Race 1 caused destruction of the most widely cultivated variety during the 20th century, Gros Michel [4,5]. This first Fusarium wilt epidemic could only be brought under control after the 1950s by the deployment of a resistant cultivar called Cavendish, which currently represents 99% of worldwide banana exports and more than 40% of global banana production [6–9]. The "return of the banana Fusarium wilt calamity" occurred in the early 1990s with outbreaks in Indonesia, Malaysia, and Taiwan, of a new Foc race called "tropical race 4" (TR4) [2,10]. This race affects Cavendish banana cultivars, as well as the other cultivars susceptible to Race 1 [2,11,12]. The resilience of Foc due to the production of survival structures called chlamydospores, and the acceleration of global trade during the last decades favoured the spread of the pathogen. Within the last two decades it has rapidly spread from its origin in Asia to major banana growing regions in the Middle East, Africa and Latin America [2,3,13–15].

Millions of people worldwide rely on banana both as a staple food and as a source of income [16], and hence solutions for the Foc-TR4 epidemic are crucial to avoid consequences of another wipe-out of the currently preferred cultivar. As a soilborne pathogen, Foc can be transmitted through irrigation, watercourses and floodwater, as well as field tools, vehicles, machinery, contaminated plants and soil. It can also survive in the soil for decades [4,11,17]. Once Foc is established within a field, there is no effective control methods to ensure its complete eradication [11].

Early detection and containment are crucial in response to Foc-TR4 epidemics in the absence of disease management strategies [11]. Molecular detection tools are the preferred method for disease detection and there is an abundance of assays published including PCR [18,19], quantitative qPCR [20,21], and loop-mediated isothermal amplification (LAMP), as reviewed in Roberts *et al.* [22]. The design of a reliable molecular detection assay for Foc-TR4 has faced several challenges. First molecular assays targeted single sequences or single nucleotide polymorphism (SNPs) in core genes (e.g., the intergenic spacer region (IGS)), whereas these genes are conserved among both target and non-target Fusaria [23]. Consequently, endophytic *F. oxysporum* with homology in these regions, are often identified as false positives in molecular assays designed in these core gene regions [24]. Another challenge consists of the specific identification of the Foc-TR4 lineage among the different races, VCGs, and Foc genetic clades. The banana Fusarium Wilt agent, Foc, is indeed a polyphyletic and diverse organism, composed of three genetic clades (A, B, C *sensu*

O'Donnell et al., 2004 [25]), corresponding respectively to the *Fusarium oxysporum* Species Complex (FOSC) Clades 1, 2, and 3 [26], eight subclusters (Foc-SC), more than 24 vegetative-compatibility groups (VCGs) [27] and four races based on the host specificity.

Foc TR4 isolates can cause disease to Cavendish banana in tropical conditions [28]. According to various authors, it is a synonym of the VCG 01213/16 [29,30], which has been controversially reclassified as *Fusarium odoratissimum* [31], and which can also include closely related VCG 0121 [2]. Race 4 is further divided into Subtropical Race 4 (STR4), which infects Cavendish only in subtropical conditions after stress predisposition [32]. Furthermore, Race 4 isolates can also affect varieties affected by Race 1 and 2, which makes the race concept confounding and challenges development of accurate molecular diagnostics [22].

Current Foc-TR4 molecular assays are also mostly confined to laboratory settings and rely on expensive molecular equipment and requires skilled operators to perform these assays. There is a need for a TR4 diagnostic system to be Point-of-care (POC), implying that the testing and assay be carried directly in the field and with rapid and simplified protocols. Loop-mediated isothermal amplification (LAMP) assays enable a rapid, simple, sensitive and robust detection [33,34]. LAMP machines are usually portable, and the enzyme used is often less sensitive to background inhibitors, which makes it more compatible with simplified DNA extraction methods that can be done directly from sampled plant material. Among the different LAMP assays developed for detecting Foc-TR4, some are unable to distinguish between TR4 and STR4 [19,35] or generate false positives [36]. Conversely, the LAMP assay developed by Ordóñez et al. [37] appears to be a reliable tool for detecting Foc-TR4.

In this study, a novel Foc-TR4 LAMP assay was designed with the aim to be field deployable. We considered a large and comprehensive genomic and genetic collection of fungal strains to identify specific markers, design adequate LAMP primers, as well as to address the limitations and specificity of the existing LAMP systems. The LAMP assay was designed with a simple and cost-effective DNA extraction protocol which can be conducted by unskilled operators. We compared its diagnostic performance with existing molecular tools including the Ordóñez LAMP assay [37]. The assay successfully detected Foc-TR4 both from artificially inoculated and naturally infected banana material and could be conducted directly in field settings. The proposed system provides a useful tool for early Foc-TR4 detection during surveillance and containment.

## Materials and methods

### Genomic resources

The genomic resources used in this study included a total of 294 genomes (summarized in Table 1 and detailed in S1 Table). Among these genomes are those available in an in-house genomic database shared between CIRAD and Stellenbosch University including 118 target Foc-TR4 genomes, 47 genomes from other Foc VCGs and 2 endophytic *F. oxysporum* isolates. Additionally, 127 genomes were downloaded from the NCBI database (National Center for Biotechnology Information, Maryland, United States) or the NGDC (National Genomics Data Centre, Beijing, China). These included 31 target Foc-TR4 genomes, 16 non-target Foc genomes, 51 genomes of various other *formae speciales* of *Fusarium oxysporum*, 29 other genomes from *Fusarium* genus, and other microbial genera commonly associated with bananas. These genomic resources were available in three different formats: Illumina short read sequencing data (n = 172), genome assemblies that had been sequenced using the Oxford Nanopore Technology (ONT) (n = 21), and genome assemblies that were publicly available (n = 101).

### Identification of a target genomic region

The K-mer elimination by cross-reference (KEC) method v.1.0 [38] (Step 1 in Fig 1) was used to identify target specific candidate sequences. This approach is fed by genome assemblies that are user-defined as target or non-target genomes.

 

**Table 1. Summary of the genomic resources and DNA from microbial isolates used in this study.**

| Species | Clade/ Foc-SCᵃ | VCG | Race | Genomic Resources | | | | TOTAL DNA Samples |
|---|---|---|---|---|---|---|---|---|
| | | | | Illumina Reads | ONT Assembly | Public Assembly | TOTAL Genomes | |
| F. oxysporum f. sp. cubense | A/1 | 0120 | STR4 | 4 | 1 | 1 | **6** | **6** |
| F. oxysporum f. sp. cubense | A/1 | 0120/15 | STR4 | 5 | 0 | 0 | **5** | **4** |
| F. oxysporum f. sp. cubense | A/1 | 01210 | STR4 | 7 | 0 | 0 | **7** | **1** |
| F. oxysporum f. sp. cubense | A/1 | 01219 | STR4 | 1 | 0 | 0 | **1** | **1** |
| F. oxysporum f. sp. cubense | A/1 | 0122 | STR4 | 1 | 1 | 0 | **2** | **2** |
| F. oxysporum f. sp. cubense | A/1 | 0129/11 | STR4 | 1 | 0 | 0 | **1** | **2** |
| F. oxysporum f. sp. cubense | A/1 | 01215 | STR4 | 1 | 0 | 0 | **1** | **1** |
| F. oxysporum f. sp. cubense | A/1 | 01227 | | 0 | 0 | 0 | **0** | **1** |
| F. oxysporum f. sp. cubense | A/1 | 0126 | STR4 | 1 | 0 | 0 | **1** | **1** |
| F. oxysporum f. sp. cubense | A/1 | 0129 | STR4 | 1 | 0 | 0 | **1** | **1** |
| F. oxysporum f. sp. cubense | A/1 | SMV 1 | UNK | 0 | 0 | 0 | **0** | **1** |
| F. oxysporum f. sp. cubense | A/1 | SMV 2 | UNK | 0 | 0 | 0 | **0** | **1** |
| F. oxysporum f. sp. cubense | A/2 | 0121 | Race 4 | 1 | 5 | 0 | **6** | **9** |
| F. oxysporum f. sp. cubense | A/2 | 01213/16 | TR4 | 126 | 14 | 8 | **148** | **46** |
| F. oxysporum f. sp. cubense | A/2 | SMV 3 | UNK | 0 | 0 | 0 | **0** | **1** |
| F. oxysporum f. sp. cubense | B/4 | 01212 | Race 1 | 1 | 0 | 0 | **1** | **1** |
| F. oxysporum f. sp. cubense | B/4 | 01220 | Race 1 | 2 | 0 | 1 | **3** | **1** |
| F. oxysporum f. sp. cubense | B/4 | 01222 | Race 1 | 1 | 0 | 0 | **1** | **1** |
| F. oxysporum f. sp. cubense | B/4 | 0124 | Race 1 | 3 | 0 | 1 | **4** | **3** |
| F. oxysporum f. sp. cubense | B/4 | 0124/25 | Race 1 | 0 | 0 | 0 | **0** | **1** |
| F. oxysporum f. sp. cubense | B/4 | 0124/5 | Race 1 | 1 | 0 | 0 | **1** | **2** |
| F. oxysporum f. sp. cubense | B/4 | 0124/5/8 | Race 1 | 1 | 0 | 0 | **1** | **1** |
| F. oxysporum f. sp. cubense | B/4 | 0125 | Race 1 | 1 | 0 | 1 | **2** | **1** |
| F. oxysporum f. sp. cubense | B/4 | 0128 | Race 1 | 1 | 0 | 0 | **1** | **3** |
| F. oxysporum f. sp. cubense | B/4 | UNK | Race 1 | 0 | 0 | 0 | **0** | **3** |
| F. oxysporum f. sp. cubense | B/4 | 01228 | UNK | 0 | 0 | 0 | **0** | **1** |
| F. oxysporum f. sp. cubense | B/4 | 01229 | UNK | 0 | 0 | 0 | **0** | **1** |
| F. oxysporum f. sp. cubense | B/4 | 01232 | UNK | 0 | 0 | 0 | **0** | **1** |
| F. oxysporum f. sp. cubense | B/4 | 01233 | UNK | 0 | 0 | 0 | **0** | **1** |
| F. oxysporum f. sp. cubense | B/4 | 01236 | UNK | 0 | 0 | 0 | **0** | **1** |
| F. oxysporum f. sp. cubense | B/4 | 0124/22 | UNK | 1 | 0 | 0 | **1** | **0** |
| F. oxysporum f. sp. cubense | B/4 | 0128/20 | UNK | 0 | 0 | 0 | **0** | **1** |
| F. oxysporum f. sp. cubense | B/4 | SMV 4 | UNK | 0 | 0 | 0 | **0** | **1** |
| F. oxysporum f. sp. cubense | B/4 | SMV 5 | UNK | 0 | 0 | 0 | **0** | **1** |
| F. oxysporum f. sp. cubense | B/4 | SMV 6 | UNK | 0 | 0 | 0 | **0** | **1** |
| F. oxysporum f. sp. cubense | B/5 | 01214 | UNK | 1 | 0 | 0 | **1** | **5** |
| F. oxysporum f. sp. cubense | B/5 | 01231 | UNK | 0 | 0 | 0 | **0** | **1** |
| F. oxysporum f. sp. cubense | B/6 | 01217 | Race 1/2 | 1 | 0 | 0 | **1** | **0** |
| F. oxysporum f. sp. cubense | B/6 | 01223 | UNK | 1 | 0 | 0 | **1** | **0** |
| F. oxysporum f. sp. cubense | B/6 | 01224 | UNK | 1 | 0 | 0 | **1** | **1** |
| F. oxysporum f. sp. cubense | B/6 | 0123 | Race 1/2 | 1 | 0 | 0 | **1** | **2** |
| F. oxysporum f. sp. cubense | B/7 | 01218 | UNK | 1 | 0 | 0 | **1** | **1** |

*(Continued)*

**Table 1.** (Continued)

| Species | Clade/ Foc-SC[a] | VCG | Race | Genomic Resources | | | | TOTAL DNA Samples |
|---|---|---|---|---|---|---|---|---|
| | | | | Illumina Reads | ONT Assembly | Public Assembly | TOTAL Genomes | |
| *F. oxysporum f. sp. cubense* | B/7 | 01234 | UNK | 0 | 0 | 0 | **0** | 1 |
| *F. oxysporum f. sp. cubense* | B/7 | 01235 | UNK | 0 | 0 | 0 | **0** | 1 |
| *F. oxysporum f. sp. cubense* | B/7 | SMV 10 | UNK | 0 | 0 | 0 | **0** | 1 |
| *F. oxysporum f. sp. cubense* | B/8 | 01221 | Race 1/2 | 1 | 0 | 0 | **1** | 1 |
| *F. oxysporum f. sp. cubense* | B/8 | 01230 | UNK | 0 | 0 | 0 | **0** | 1 |
| *F. oxysporum f. sp. cubense* | C/3 | 01226 | UNK | 0 | 0 | 0 | **0** | 3 |
| *F. oxysporum f. sp. cubense* | UNK | UNK | Race 1 | 1 | 0 | 3 | **4** | 5 |
| *F. oxysporum f. sp. cubense* | UNK | UNK | STR4 | 12 | 0 | 0 | **12** | 0 |
| *F. oxysporum f. sp. cubense* | UNK | UNK | UNK | 0 | 0 | 5 | **5** | 0 |
| *F. oxysporum* | | | Banana endophyte | 2 | 0 | 0 | **2** | 3 |
| *F.oxysporum*, other *formae speciales* [b] | | | | 0 | **29** | 30 | **12** | 0 |
| *F. oxysporum*, others [c] | | | | 0 | 0 | 22 | **22** | 12 |
| *Fusarium*, other species [d] | | | | 0 | 29 | 29 | 3 | 0 |
| *Nectria sp.* FW16.1 | | | | 0 | 0 | 1 | **1** | 0 |
| *Pseudocercospora fijiensis* | | | | 0 | 0 | 0 | **0** | 3 |
| *Ralstonia solanacearum* | | | | 0 | 0 | 0 | **0** | 1 |
| **Total Target (TR4)** | | | | **126** | **14** | **8** | **148** | **46** |
| **Total Non-Target** | | | | **46** | **7** | **93** | **146** | **115** |
| **Total** | | | | **172** | **21** | **101** | **294** | **161** |

The data type used is indicated as Illumina short read genomes, ONT-based assemblies, or public assemblies (from publicly available genomes). All F. *oxysporum* f. sp. other than *cubense* were not pathogenic on banana.

[a]The Clade and VCG classification is based on Mostert *et al*., 2022 [27]. SMV: Single-Member VCG. UNK: Unknown.

[b]The other *formae speciales* of *Fusarium oxysporum* tested were: *canariensis, cepae, ciceris, conglutinans, cucumerinum, cyclamini, dianthi, fragariae, koae, lini, lycopersici, radicis-lycopersici, radicis-vanillae, rapae, strigae, vanillae, vasinfectum, spinaciae*

[c] *F.oxysporum* strains which were not assigned to a specific *forma specialis*.

[d]The other *Fusarium* species tested included F.*agapanthi, F. avenaceum, F. falciforme, F. fujikuroi, F. graminearum, F. haematococcum, F. langsethiae, F. louisianense, F. metavorans, F. neocosmosporiellum, F. pseudoanthophilum, F. pseudonygamai, F. ramigenum, F. redolens, F. solani, F. udum, F. vanettenii, F. venenatum, F. verticillioides.*

In addition to the 21 ONT-based in-house assemblies [39,40], and the 101 publicly available genomes, all the Illumina short read genomes (172) were assembled to maximise the number of input genomes. The assembler ABySS **v.2** [41] software was used with an automated optimised best K-mer selection through the *ABYSS_launch.py* workflow, written by Florian Charriat (https://github.com/FlorianCHA/script_paper). Importantly, these assemblies were repeat-masked using RepeatMasker (http://repeatmasker.org/).

The ONT-based in-house assembly of CAV807 ([39], available upon request) was chosen as the main target reference genome (master) on the KEC method, using the "pool" feature with the rest of target genomes. A combination of parameters was tested to do KEC computations (test details included on S2 Table) with a minimum output sequence size range from 60 to 500 bp, K-mer size ranges from 13 to 19, different input populations, and including or excluding reverse complement sequences, all of which produced different output candidate sequence lists. All these sequences were then merged on a single file, and were used as reference sequences on the following steps (Steps 24 in Fig 1).

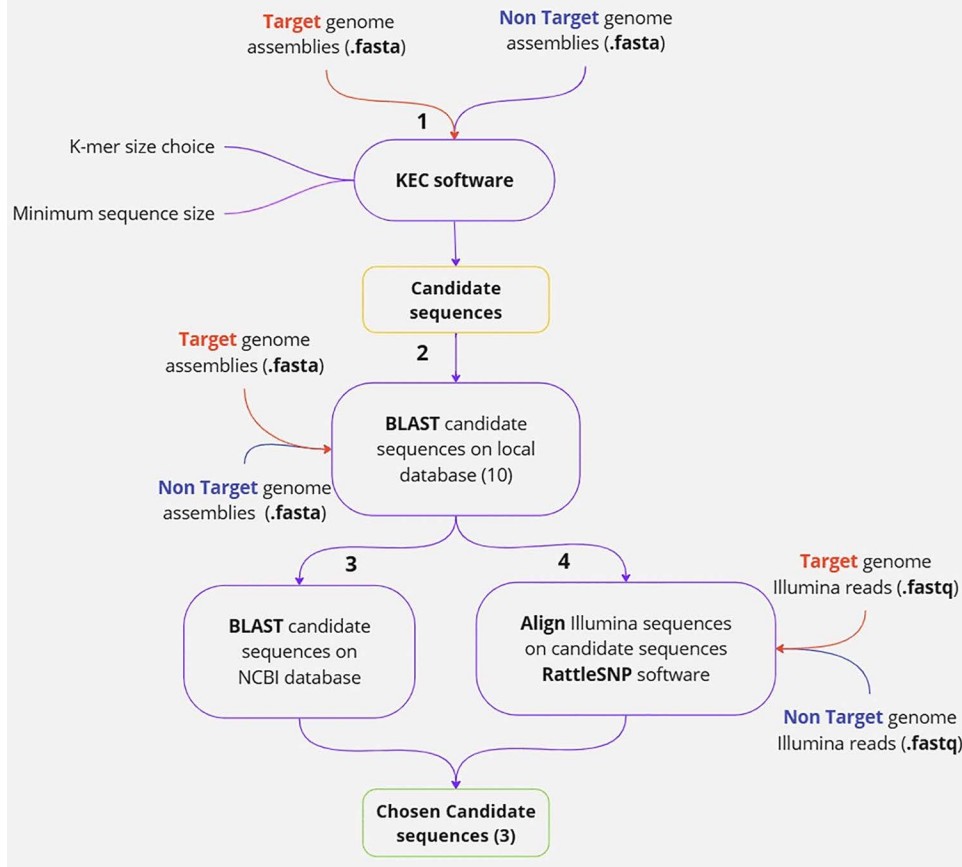

**Fig 1. Pipeline followed to find TR4-specific sequences. Step 1** assemblies of target and not-target genomes were used for the KEC (K-mer elimination by cross-reference) method described by Beran et al. [38] to obtain an initial list of target specific candidate sequences. These sequences were queried (**Step 2**) in an internal BLAST to double-check the presence/absence on non-target genomes. The remaining candidate sequences were filtered in two ways. Firstly, based on the results obtained when a query was done with BLAST on the NCBI online database (**Step 3**), whilst the second (**Step 4**) aligned Illumina reads on the candidate sequences with RattleSNP software (https://github.com/sravel/RattleSNP), and computed coverage of query reads on sequence tested. Figure created with Miro™.

To verify that the output sequences were present on the target genomes while absent from non-target genomes, an internal BLASTN query (Step 2, Fig 1) was done for all the candidate sequences on all the available genomes; candidates that had high BLAST hits on non-targeted genomes (more than 90% identity of full queried sequence) were discarded. The best sequence candidates were also queried against fungal genomes (Whole-Genome shotgun contigs database version for Fungi, taxid 4751) on the NCBI database (step 3, Fig 1) (https://blast.ncbi.nlm.nih.gov/Blast.cgi). The candidate sequences were regarded as good targets for primer design if (i) they got a hit on all the available TR4 genomes on NCBI (named "*Fusarium oxysporum* f. sp. *cubense* TR4", or "*F.odoratissimum*"), and (ii) they got the minimum hit on non-TR4 genomes. As a reference, internal BLASTN and NCBI database queries were performed in parallel using the sequence targeted in the Ordóñez LAMP assay [37], which was the latest published LAMP TR4-diagnostic region at the time of this study.

Additionally, the genomic Illumina reads of all genomes available in this format were mapped (Step 4 Fig 1) on the candidate sequences used as references, using the workflow RattleSNP (https://github.com/sravel/RattleSNP). Mapping 150 bp reads on such a short reference sequence may result in misalignments. To avoid this, the candidate sequences

were extended 1000 bp up and downstream using the unmasked version of the ONT-based CAV807 assembly, before the alignment step. The sequences which had the highest percentages of coverage among all target Illumina genomes (TR4), and lowest coverage on non-targeted genomes, were selected.

## PCR primer design and initial screening of selected candidate sequences

Initial screening of best performing candidate sequences was done either in PCR or Real-time quantitative PCR (qPCR). Primers were designed using Primer 3 (https://primer3.ut.ee/), and assays were performed on a bCUBE machine (Hyris Ltd, London, United Kingdom), adjusting the amplification protocol for each case primer system. For PCRs, the Promega GoTaq® G2 MasterMix (Promega, Wisconsin, United States) kit was used, and PCR products were assessed on a 1.5% Agarose Gel, whilst for qPCR the SYBR™ Select Master Mix qPCR (Thermo Fischer, Waltham, Massachusetts, United States) was used.

## LAMP primer design and assay conditions

LAMP primers were designed using the Primer explorer V5 software (primerexplorer.jp/lampv5e/index.html). When software did not yield Loop primers by default settings, design was done manually. A similar approach was used for the design of Swarm primers for which design option is not included in the software. LAMP reactions were performed on the portable Genie II machine (Optigene, Blatchford, United Kingdom), in a total reaction volume of 25 μL, containing 15 μL ISO-DR004 Isothermal Mastermix (OptiGene, Horsham, UK), 2.5 μL of pre-mixed Primer Mix, (resulting in final concentrations of 0.2 μM of each F3 and B3 primers, 0.8 μM of each FIP and BIP primers, 0.4 μM of each loop primer), and 5 μL of template DNA. Note that within the specificity tests, 1 μL of template DNA was used instead of 5 μL. LAMP reactions were run at 65°C for 30 min, followed by a 6 min annealing step with temperatures varying from 98°C to 80°C at a speed of 0.05°C per second. Amplification curves associated with a "Time to Result value" (TTR values) and annealing temperature (*Ta*) peaks were generated and recorded during the LAMP reaction. A sample was considered positive if (i) the average TTR of the replicates of a sample was below the established minimum TTR threshold (30:00), (ii) if the average *Ta* peak was ± 1 °C from the reference *Ta* (relative to each LAMP system), and (iii) if the fluorescence of the *Ta* peak reached a minimum threshold value of 1500. By convention, a value of 31:00 (just above the time limit) was assigned to a negative result (TTR = 00:00) for the calculations and graphical representations (as in [42]). In the case of sensitivity tests, the limit of detection (LOD) was considered as the minimum dilution of DNA at which the mean of the replicates could produce a positive result at a 95% of confidence level.

## Reference assays

The Ordóñez LAMP assay [37] was used with the LAMP conditions described above and the program described in the publication. Additionally Two TR4-specific qPCRs systems were considered as references: (i) The hydrolysis probe-based real-time assay developed by Aguayo [20], which is the current reference method in France by the French Agency for Food, Environmental and Occupational Health and Safety, ANSES [43], along with the Promega GoTaq™ Probe qPCR Master Mix (Promega, Wisconsin, United States); and (ii) the SYBR™-based real-time assay [21] along with the SYBR™ Select Master Mix qPCR (Thermo Fischer, Waltham, Massachusetts, United States). A sample was considered positive when the mean Threshold Cycle (Ct) over all replicates was below a threshold of 35 cycles.

## Specificity testing

A set of 161 DNA samples was used for specificity testing in the present study; including 46 target Foc-TR4 isolates and 82 non-target Foc isolates representing intraspecies diversity (S3 Table). In addition, other non-target isolates included *F. oxysporum* endophytes and other *F. oxysporum formae speciales* (n = 25), other *Fusarium* spp. (n = 5), and pathogens of

banana (3 isolates of *Pseudocercospora fijiensis,* causing the black Sigatoka; and one isolate of *Ralstonia solanacearum,* causing the Moko disease). The Moko-inducing isolate was grown onto tetrazolium chloride medium (TZC) at 28 °C [44]. Bacterial cultures grown overnight were used for DNA extraction using a DNeasy Blood and tissues kit (Qiagen, Courtaboeuf, France). Fungal cultures were grown on potato dextrose agar (PDA) for seven days at 25°C, then transferred to a liquid medium (Nitrate medium; Yeast Nitrogen Base: 1.7 g L$^{-1}$, Sucrose: 30 g L$^{-1}$, KNO$_3$: 10.11 g L$^{-1}$; M. de Saint & M. Rep, personal comm. 2020) in an Erlenmeyer flask for five to seven days. The mycelia were filtered through sterile cheesecloth by rinsing with sterile water. Mycelia of each isolate was then ground with mortar and pestle in liquid nitrogen prior to DNA extractions performed following an in-house protocol (S1 Methods). DNA quantity and quality were assessed on a Nanodrop™ spectrophotometer (Thermo Fischer, Waltham, Massachusetts, United States). DNA concentration was adjusted to 1−2 ng µL$^{-1}$ before it was tested with optimised LAMP assays. The rest of extractions were performed in Stellenbosch University following protocols detailed earlier [45,46].

## Sensitivity testing

Different DNA samples were prepared for sensitivity tests including pure plasmid DNA samples, plasmid spiked plant material, as well as plant material artificially inoculated with conidial suspensions.

### Sensitivity based on plasmid dilution series

A plasmid containing the selected DNA sequence targeted by the LAMP primers was ordered from Eurofins (Eurofins, Luxembourg) (S1 Fig). This was initially resuspended in TE buffer (Sigma-Aldrich, Massachusetts, United States) as recommended by the manufacturer, and dilution series were prepared in water (10$^8$ copies µL$^{-1}$ to 10$^0$ copies µL$^{-1}$).

### Sensitivity based on plasmid-spiked plant material

Plasmid-spiked plant tissue samples were prepared by first grinding healthy plant tissue from corm and pseudostem in liquid nitrogen, after which approximately 100 mg of healthy tissue were collected in Eppendorf tubes. To spike the samples, 10 µL of each plasmid dilution were mixed with the ground material. DNA was then extracted with the DNeasy Plant Kit protocol as per manufacturer's instructions (Qiagen, Hilden, Germany), but eluted in a final volume of 100 µL, which was used to calculate the final concentration of plasmid in each sample (from 10$^5$ conidia µL$^{-1}$, downto 0 conidia µL$^{-1}$).

### Sensitivity based on plant material inoculated with conidial suspensions

The Foc-TR4 isolate MAY0001 was used to prepare an initial conidial suspension. Conidial concentrations (10$^7$ conidia mL$^{-1}$ downto 10$^0$ conidia mL$^{-1}$) were adjusted with haemocytometer (KOVA, California, United States) visualised under a light microscope. Each conidial dilution was mixed with about 100 mg of ground healthy plant tissue from corm and pseudostem, and then centrifuged at 14 rpm for 5 min. DNA extraction was performed on the pellet using the DNeasy Plant Kit (Qiagen, Hilden, Germany). Extractions were done in duplicate for each prepared concentration (from 10$^5$ conidia µL$^{-1}$, to 0 conidia µL$^{-1}$).

## Detection from artificially inoculated material

Conidial suspensions of MAY0001 were prepared as described above. Three-month old tissue culture-derived banana plants were used, which were previously hardened in the CIRAD-PHIM glasshouse (Montpellier, France). Infections were conducted by pouring 20 ml conidial suspensions at a concentration of 1x10$^6$ conidia mL$^{-1}$ around the base of plant. After ten weeks of incubation (30°C day/ 26°C night, 90% RH, 2h of additional lighting (300W m$^{-2}$) to ensure 12h daylight during winter), infection was assessed by cross-sections of the corm. Where infection was evident including reddish brown discoloration of the vascular tissues and rhizomes, tissue was collected and frozen at −20°C. Tissue was then ground to

a powder in liquid nitrogen and transferred to Eppendorf™ tubes (Eppendorf, Hamburg, Germany). About 100 mg of plant material from each sampled plant were used for DNA extraction with the DNeasy Plant Kit following the instructions of the manufacturer (Qiagen, Hilden, Germany). DNA quantity was measured with Nanodrop. The sensitivity of the optimised LAMP assay to detect Foc-TR4 DNA was compared to the mentioned published assays.

### In-field deployment and detection from naturally infested plant material

The optimised LAMP assay was deployed on site during a field survey in Java, Indonesia, organized in collaboration with the local institute BRIN (National Research and Innovation Agency). The survey was conducted in the Cianjur regency, at four different plots in the Cikalongkulon and Mande districts where Foc-TR4 cases were previously reported. Thirty-two plants of different banana cultivars, including both Cavendish and non-Cavendish cultivars of various genomic formulas (detailed in S7 Table) were sampled, making a total of 70 samples collected from fruit, core of pseudostem, outer pseudostem, corm, and peduncles, which were analysed both with the LAMP assay optimised in this study, the published LAMP [37] and the Matthews qPCR [21]. Additionally, eight pseudostem samples collected from Comoros (Grande Comore) and Vietnam and sent to Stellenbosch University, were also analysed.

A rapid DNA extraction method was modified from Robène *et al* [42], for extraction from collected pseudostem tissues: About 250 mg of pseudostem tissue was transferred to a Bioreba extraction bag containing 1 ml 0.5 M NaOH including 2% PVP added, and was then ground with hand grinder (Bioreba, Reinach, Switzerland). After grinding, 5 µL of the resulting homogenate liquid was collected and diluted into 195 µL 100 mM Tris (pH 8.0). This homogenate was immediately analysed in LAMP assays. In parallel, the Mathews qPCR [21] was performed on the duplicate samples extracted with a magnetic bead extraction kit according to the supplier's protocol (Sera-Mag Select, cytiva, Marlborough, USA).

### Statistical analysis for the test comparisons

The results of the LAMP assays on plant material were compared with each other and with the qPCR results by the Cohen's kappa index and the McNemar's test [47]. Cohen's kappa index was interpreted as in [48] to evaluate the agreement between techniques: Kappa ≤ 0 corresponds to "no agreement", 0.01–0.20 is "none to slight", 0.21–0.40 is "fair", 0.41–0.60 is "moderate", 0.61–0.80 is "substantial", and 0.81–1.00 is "near-perfect". Additionally, different diagnostic parameters were calculated, such as the Sensitivity, Specificity, Positive Predictive Value (PPV), Negative Predictive Value (NPV), and Accuracy as in [49]. A McNemar's p-value below 0.05 was taken as evidence to reject the null hypothesis of equal marginal proportions, indicating a statistically significant difference in detection outcomes between the two methods compared. This would reflect an imbalance in discordant results, i.e., a real shift in the numbers of false positives versus false negatives, which in turn underlies any observed differences in diagnostic parameters such as sensitivity and specificity.

### Ethics statement

Field samplings were done in Java, Indonesia, as granted by the National Research and Innovation Agency BRIN (Badan Riset dan Inovasi Nasional, https://www.brin.go.id).

## Results

### *In silico* identification and validation of candidate target region

A total of 1170 candidate target sequences were yielded with the KEC method. The internal BLASTN filtering step (by querying the candidate sequences against the same database) (Step 2, Fig 1) was found a mandatory step, as many candidates were detected both in target and non-target genomes (generally phylogenetically close lineages). Duplicated candidate sequences were also removed, selecting 30 candidate sequences to analyse (details in S2 Table). The list was

further reduced to 10 candidate sequences when the amount of non-target genomes was increased in the analysis and sequences showing multiple copies over the target regions were discarded. After the *in silico* specificity testing done by BLAST query on NCBI database on Whole Genome Contigs for all Fungi (taxid 4751) (Step 3, Fig 1), three candidates were finally selected including Candidate 5 (709 bp), Candidate 23 (417 bp) and Candidate 27 (515 bp) located in contigs VMNF01000005.1, VMNF01000003.1 and VMNF01000004.1 of UK0001 reference genome respectively (contigs 12, 3, and 2 on the CAV807 reference used) (Fig 2). All three candidates were identified in intergenic regions, with Candidate 5 being close to a putative gene. All appeared to be target-specific, as their sequence identities to non-target genome sequences did not exceed 90% in the internal database (S4 Table). In external NCBI BLAST analysis (Step 3, Fig 1), three candidates showed some hits with non-target taxa, all with sequence identities below 84% and with partial regions of the queried sequence (Details in S4 Table). Candidate 5 had a non-specific hit on an *Ascodesmis sphaerospora* strain, Candidate 23 showed non-specific BLAST hits with *F. oxysporum* Fo10, an Australian endophyte isolated from soil (Constantin et al., 2021), and on *F. fujikuroi*. Candidate 27 had hits with a *F. proliferatum* genome.

None of the candidate sequences had BLAST-positive hits on the full number of target Foc-TR4 Illumina assemblies used, leading us to a further mapping exploration using the RattleSNP workflow (Step 4, Fig 1). This approach considers alignment coverage and depth of Illumina sequence reads, rather than sequence identity on assemblies, which allows to explore sequences that had been discarded during assembly process. From all 120 Foc-TR4 Illumina genomes, nine and seven genomes were not found to map at all on the candidates 5 and 27, respectively (Table 2). However, since the missing genome lists of these two markers were complementary to each other, these two markers were kept designing a potential duplex. Candidate 23 on the other hand, was fully inclusive with all 120 Foc-TR4 genomes, although mapping depth was low in some regions of the candidate sequence.

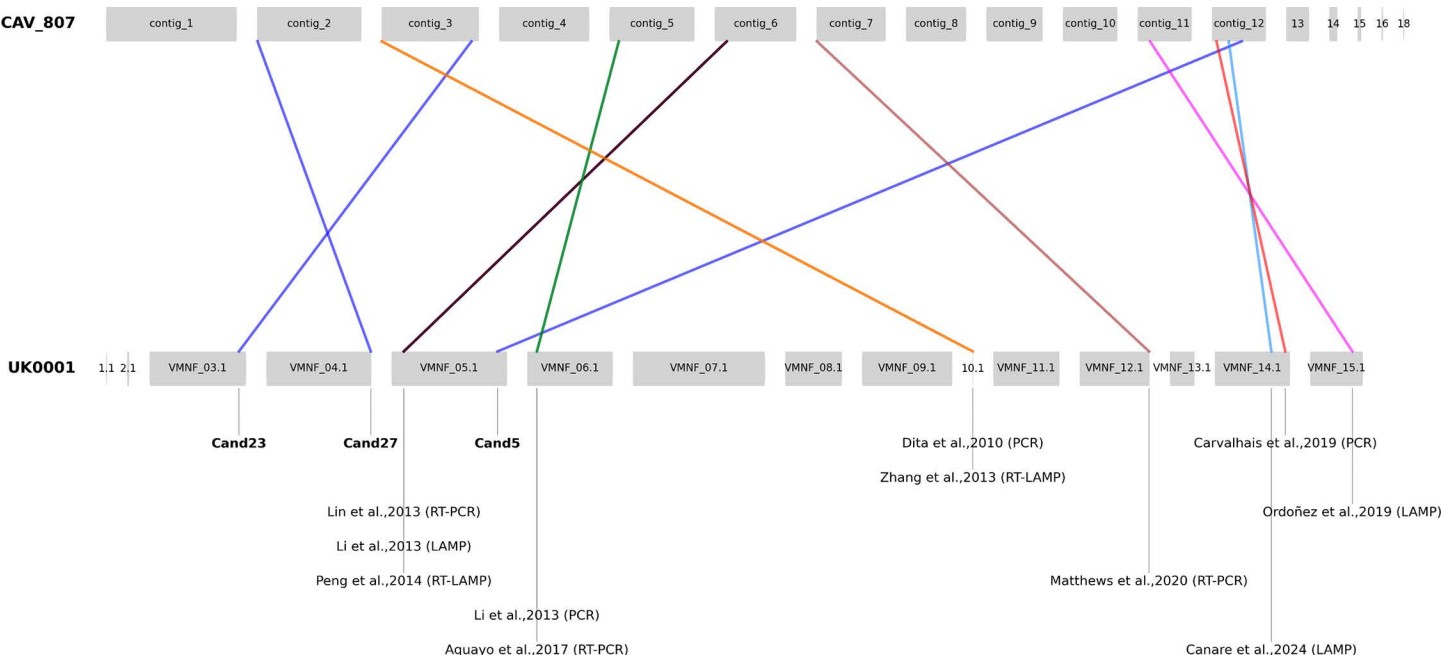

**Fig 2. Genomic locations of the amplicons of previously published Foc-TR4 diagnostic systems used on the genomes of UK0001 and CAV807.** The three amplicons of the three candidates described in the current publication are written in bold. The chromosome names of the UK0001 have been simplified to fit in the Figure, being originally "VMNF010000…".

**Table 2.** *In silico* specificity of candidate Foc-TR4 target sequence regions and LAMP amplicons of those previously published.

| Marker Region | Reference sequence used for alignment / BLAST queries | | | | | | | | | Genomes used | Internal BLAST results of reference sequences on all assembled genomes (Identity %) | | | | Mapping results of Illumina reads on the Reference Sequences | | |
| | AT[a] | ST[a] | L[a] | MC[a] | Mask.[a] | Loc.[a] | Position in CAV807 / UK0001 reference genomes | | | Status | NTBH[b] | 100% | 95-100% | 90-95% | GFM[c] | GPM[c] | GNM[c] |
| | | | | | | | Contig | Start | End | | | | | | | | |
| Cand5 | LAMP | Amplicon | 214 | No | No | Inter | contig_12 / VMNF01000005.1 | 1486738 / 5279483 | 1486951 / 5279270 | Target | No | 133 | 0 | 0 | 111 | 0 | 9 |
| | | | | | | | | | | Non-Target | | 0 | 0 | 0 | 0 | 2 | 46 |
| Cand23 | LAMP | Amplicon | 247 | No | No | Inter | contig_3 / VMNF01000003.1 | 4477508 / 4455382 | 4477754 / 4455628 | Target | No | 119 | 0 | 0 | 118 | 2 | 0 |
| | | | | | | | | | | Non-Target | | 0 | 0 | 0 | 0 | 10 | 38 |
| Cand27 | LAMP | Amplicon | 234 | No | No | Inter | contig_2 / VMNF01000004.1 | 32096 / 5203206 | 32329 / 5202973 | Target | Yes | 133 | 0 | 0 | 112 | 1 | 7 |
| | | | | | | | | | | Non-Target | | 0 | 0 | 0 | 0 | 7 | 41 |
| Ordoñez et al. (2019) | LAMP | Amplicon | 214 | No | No | Inter | contig_11 / VMNF01000015.1 | 623539 / 2103661 | 623752 / 2103448 | Target | Yes | 140 | 0 | 0 | 118 | 0 | 2 |
| | | | | | | | | | | Non-Target | | 0 | 1 | 0 | 1 | 3 | 44 |
| Canare et al. (2024) | LAMP | Amplicon | 213 | Yes | No | Inter | contig_12 / VMNF01000014.1 | 842769 / 2814708 | 842981 / 2814920 | Target | Yes | 142 | 0 | 2 | 120 | 0 | 0 |
| | | | | | | | | | | Non-Target | | 17 | 2 | 17 | 17 | 5 | 26 |
| Li et al. (2013) | LAMP | Amplicon | 213 | No | Yes | Genic | contig_6 / VMNF01000005.1 | 603139 / 613080 | 603351 / 613292 | Target | Yes | 143 | 0 | 0 | 120 | 0 | 0 |
| | | | | | | | | | | Non-Target | | 3 | 25 | 0 | 25 | 20 | 3 |
| Zhang et al. (2013) | RT-LAMP | Amplicon | 472 | Yes | No | Inter | contig_3 / VMNF01000010.1 | 5266 / 17344 | 5737 / 16873 | Target | Yes | 27 | 4 | 0 | 120 | 0 | 0 |
| | | | | | | | | | | Non-Target | | 0 | 22 | 3 | 48 | 0 | 0 |
| Peng et al. (2014) | RT-LAMP | Amplicon | 217 | No | Yes | Genic | contig_6 / VMNF01000005.1 | 603149 / 613090 | 603365 / 613306 | Target | Yes | 143 | 0 | 0 | 120 | 0 | 0 |
| | | | | | | | | | | Non-Target | | 3 | 25 | 0 | 27 | 19 | 2 |

[a] Summary of the genome locations, BLAST and sequence alignment results. AT: Assay Type; ST: Sequence Type; L: length (bp); MC: Multiple Copies; Mask.: Located in repeat-masked regions; Loc.: location, genic or intergenic (inter).

[b] NTBH: Any Non-Target BLAST hit (any size).

[c] GFM: Genomes fully mapped (100% coverage); GPM: genomes partially mapped; GNM: Genomes not mapped (0% coverage).

Coverage over non-target genomes was below 10% for all the three candidate regions (Full details on S4 Table) except for CAV_36117 (Clade B, Foc-SC04), which mapped with a 12.6% and 43.7% coverage on the sequences of Candidate 5 and 23 respectively, and CAV_612 (Clade A, Foc-SC01), mapping with a 55.1% of coverage on Candidate 27.

### *In silico* specificity analysis of candidate regions vs. previously published regions

The approach of internal BLAST and Illumina alignments (Steps 2 and 4 in Fig 1), initially tested on the new candidates, was also applied on already published Foc-TR4 diagnostics assays by selecting the amplicon sequences of these systems (Table 2, details on S4 Table). The genomic locations of the different markers on the genomes UK0001 and CAV807 are shown on Fig 2. Considering LAMP systems, none of the published regions were found to be fully specific to our genome panel.

The Ordóñez LAMP amplicon [37] showed the lowest amount of false positive detections, but had hits of 95–100% sequence identity to eight non-target genomes when tested on the internal BLAST dataset, most notably sequence identity of 97.2% to isolate CAV 225, an *F. oxysporum* isolated soil in banana plantation in South Africa. In addition, the Ordóñez system [37] did not have any sequence alignment with two Foc-TR4 Illumina genomes: SHN19, and FJ41 from China. Considering the other four LAMP systems studied (Table 2), they all showed good inclusivity, by mapping with all 120 Illumina TR4 genomes, but were found to be less specific than Ordóñez *et al.* (2019). The Canare *et al.* 2024 marker [50] showed 100% sequence identity with 17 non-targeted assembled genomes, most of which were STR4, VCG 0121, and three Australian unknown VCG Foc isolates. It also showed full alignment with 17 non-target Illumina genomes including various known Foc VCGs (0120, 0121, 0122, 0126, 01215, 01219). The LAMP target reported from [36], designed based on the IGS gene regions [18], showed a 100% alignment with 48 non-target Illumina genomes, and a BLAST identity between 95–100% in 40 non-target alignments, including Foc isolates identified as STR4, Race 1 and isolates in other Foc VCGs. The LAMP amplicons of Li *et al.* (2013), and Peng *et al.* (2014) [19,35], located nearby close to the amplicon of the RT PCR Lin *et al.* (2013) [51], were found in full alignment within 25 and 27 non-target Illumina genomes, respectively. When analysed in BLAST both gene targets showed a 100% identity within three VCG 0121 genome, and showed a 95–100% identity with 25 non-target genomes.

We also benchmarked other non-LAMP system amplicons (S4 Table). The qPCR amplicon published by Matthews *et al.* 2020 [21], showed the best inclusivity-exclusivity, although it was found to be located in a repeats-rich region of the genome (thus masked by RepeatMasker). The Aguayo qPCR amplicon [20] showed a 100% BLAST identity on the three VCG 0121 genomes (TWN0005, TWN0004, and TWN0003), and high coverage on 47 non-target genomes, mostly with Race 4 isolates.

### Primer design and initial validation

A first screening was done on a subset of the specificity DNA panel, confirming that the best candidates were Candidate 5, 23 and 27 (respectively named Cand-5, Cand-23, and Cand-27). LAMP primers were then designed on the three candidates to be tested in the specificity panel.

Due to the limited size of the highly specific part of the Cand-23 fragment, only LAMP primer sets including FIP, BIP, F3, and B3 could be designed using the primer explorer software, and a Loop and two swarm primers [52] designed manually, were included in the final version of the Candidate 23 LAMP assay (Table 3, Fig 3).

### LAMP Primer Specificity

The Cand-23 LAMP assay was the most specific one (Table 4), only amplifying true positives and not amplifying any non-target isolates among the 161 strains tested. Cand-5 LAMP assay (121 strains tested) amplified 3 false positives and 7 false negatives, which were some of the genomes found absent *in silico*. This would confirm that their absence was genuine rather than due to sequencing artefacts. Cand-27 showed one false negative (62 strains tested).

Table 3. Primer sequences and concentration used for Cand-23 LAMP assay.

| Primer | Sequence | FC[a] |
|---|---|---|
| FIP-cand23 | AAGCACTTATAAAACCCGGCCTCGAAGTAAACTAACTAGGGCAAC | 1.6 |
| BIP-cand23 | AAGCTGGCTTAGCCTTAAAAGCTCCTATTAAGGGCAAAGAGAC | 1.6 |
| Loop-cand23 | TTCTAATTACTTTTATTAGCACTGC | 0.8 |
| Swam_F1c-cand23 | AAGCACTTATAAAACCCGGCCTC | 0.8 |
| SwarmB1c-cand23 | AAGCTGGCTTAGCCTTAAAAGC | 0.8 |
| F3-cand23 | TTAGAGGTATAAGGAGGCTTAG | 0.4 |
| B3-cand23 | GCCTTTACTTAGTAATCTTTTACCT | 0.4 |

[a]FC: Final concentration (μM), required in the final 25 μL reaction.

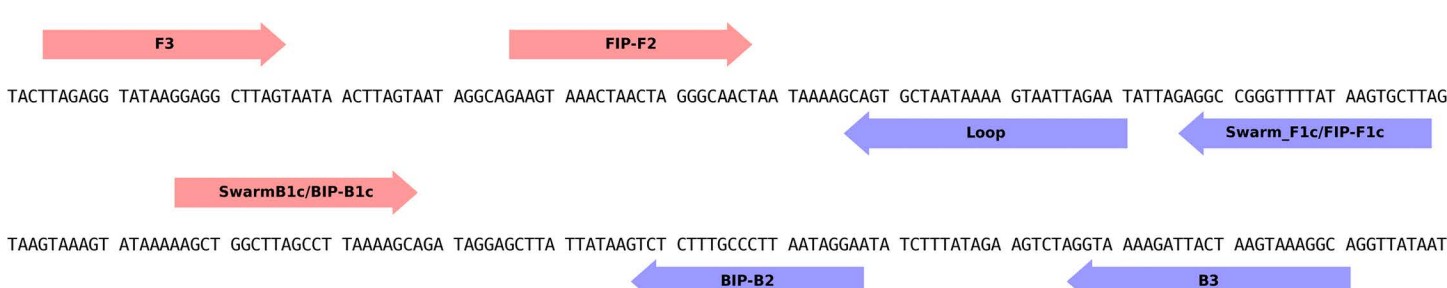

Fig 3. Position and direction of the LAMP primers on the Cand-23 region.

Therefore, the Cand-23 LAMP assay was chosen for the subsequent tests. Considering other molecular detection tools that were analysed in this study, the Matthews qPCR [21] was found to be fully specific among 104 isolates tested. The Aguayo qPCR [20] amplified all target isolates, and 15 non-target (VCG 0121) isolates. This was not surprising as it is known that this system also amplifies VCG 0121. The Ordoñez LAMP assay [37] (158 isolates tested) picked up all Foc-TR4 target isolates but also amplified 3 non-target strains: CAV 531 (endophyte), CAV 615 (VCG 01227), and Fova1185, a *Fusarium oxysporum* isolated from vanilla, respectively. Detailed results of the specificity are shown in S5 Table.

## LAMP Primer Sensitivity and optimisation

Preliminary optimisation of Cand-23 LAMP primer concentrations and combinations allowed the selection of a primer mix (PM) consisting of doubled initial standard concentrations (FIP and BIP at 1.6 μM; SwarmF1c, SwarmB1c and Loop primers at 0.8 μM; and B3 and F3 at 1.4 μM) (Table 3), as this configuration reduced the average TTR by 1 min 45 s (S2 Fig).

Sensitivity of the Cand-23 LAMP assay was tested on plasmid dilutions prepared in water (Fig 4, a) and on DNA extractions of healthy plant tissue spiked with plasmid dilutions (Fig 4, b). In both sample sets, the limit of detection (LOD) was reached consistently at $10^2$ copies μL$^{-1}$. When a volume of 5 μL per reaction of the plasmid DNA was used (blue points in the graphs), TTR averaged 18:07 min, and Annealing temperature (At) of 82.8 °C for plasmid dilutions in water, while TTR and At averaged respectively 13:41 min and 82.3 °C in plant tissue. Surprisingly, when sensitivity tests were conducted on samples were plant tissue were spiked with plasmid dilutions, a better detection limit were recorded at $10^1$ copies μL$^{-1}$ using 5 μL per reaction, that is 50 copies per reaction (average TTR 20:32 min, 82.3 °C in Fig 4, b).

**Table 4. Specificity results of the LAMP markers designed in this project (Cand-23, Cand-5, and Cand-27), compared with three current Foc-TR4 diagnostic reference methods.**

| Isolates tested | | | | | Result of different Foc-TR4 diagnostic systems | | | | | | | | | | | |
| --- | --- | --- | --- | --- | --- | --- | --- | --- | --- | --- | --- | --- | --- | --- | --- | --- |
| | | | | | Existing systems | | | | | | This study | | | | | |
| Status | Species[a] | Clade / Foc-SC[b] | VCG[a] | Race[a] | qPCR1[c] | | qPCR2[c] | | LAMP3[c] | | LAMP Cand-23 | | LAMP Cand-5 | | LAMP Cand-27 | |
| | | | | | +[d] | -[d] | +[d] | -[d] | +[d] | -[d] | +[d] | -[d] | +[d] | -[d] | +[d] | -[d] |
| Non-Target | F. oxysporum f. sp. cubense | A/1 | 0120 | STR4 | 0 | 2 | 0 | 2 | 0 | 6 | 0 | 6 | 0 | 3 | 0 | 2 |
| | | | 0120/15 | UNK | 0 | 3 | 0 | 3 | 0 | 5 | 0 | 5 | 0 | 3 | 0 | 1 |
| | | | 01210 | UNK | 0 | 1 | 0 | 1 | 0 | 1 | 0 | 1 | 0 | 1 | 0 | 0 |
| | | | 01219 | UNK | 0 | 0 | 0 | 0 | 0 | 1 | 0 | 1 | 0 | 1 | 0 | 1 |
| | | | 0122 | UNK | 0 | 2 | 0 | 2 | 0 | 2 | 0 | 2 | 1 | 1 | 0 | 2 |
| | | | 0129/11 | UNK | 0 | 1 | 0 | 1 | 0 | 2 | 0 | 2 | 0 | 2 | 0 | 1 |
| | | | 01215 | UNK | 0 | 0 | 0 | 0 | 0 | 1 | 0 | 1 | 0 | 1 | 0 | 1 |
| | | | 01227 | UNK | 0 | 0 | 0 | 0 | 1 | 0 | 0 | 1 | 0 | 1 | 0 | 1 |
| | | | 0126 | UNK | 0 | 1 | 0 | 1 | 0 | 1 | 0 | 1 | 0 | 1 | 0 | 1 |
| | | | 0129 | UNK | 0 | 0 | 0 | 0 | 0 | 1 | 0 | 1 | 0 | 1 | 0 | 1 |
| | | | SMV 1 | UNK | 0 | 0 | 0 | 0 | 0 | 1 | 0 | 1 | 0 | 1 | 0 | 1 |
| | | | SMV 2 | UNK | 0 | 0 | 0 | 0 | 0 | 1 | 0 | 1 | 0 | 1 | 0 | 1 |
| | | A/2 | 0121 | Race 4 | 8 | 0 | 0 | 8 | 0 | 9 | 0 | 9 | 0 | 9 | 0 | 3 |
| Target | | | 01213/16 | TR4 | 23 | 0 | 21 | 0 | 36 | 0 | 37 | 0 | 17 | 7 | 11 | 1 |
| | | | N.D. | TR4* | 9 | 0 | 9 | 0 | 9 | 0 | 9 | 0 | 0 | 0 | 0 | 0 |
| Non-Target | | | SMV 3 | UNK | 0 | 0 | 0 | 0 | 0 | 1 | 0 | 1 | 1 | 0 | 0 | 1 |
| | F. oxysporum f. sp. cubense | B/4 | 01212 | Race 1/2 | 0 | 1 | 0 | 1 | 0 | 1 | 0 | 1 | 0 | 1 | 0 | 1 |
| | | | 01220 | UNK | 0 | 0 | 0 | 0 | 0 | 1 | 0 | 1 | 0 | 1 | 0 | 1 |
| | | | 01222 | UNK | 0 | 0 | 0 | 0 | 0 | 1 | 0 | 1 | 0 | 1 | 0 | 1 |
| | | | 0124 | UNK | 0 | 2 | 0 | 2 | 0 | 3 | 0 | 3 | 0 | 3 | 0 | 1 |
| | | | 0124/25 | UNK | 0 | 1 | 0 | 1 | 0 | 1 | 0 | 1 | 0 | 0 | 0 | 0 |
| | | | 0124/5 | UNK | 0 | 0 | 0 | 0 | 0 | 2 | 0 | 2 | 0 | 2 | 0 | 2 |
| | | | 0124/5/8 | UNK | 0 | 1 | 0 | 1 | 0 | 1 | 0 | 1 | 0 | 1 | 0 | 0 |
| | | | 0128 | UNK | 0 | 1 | 0 | 1 | 0 | 3 | 0 | 3 | 0 | 3 | 0 | 2 |
| | | | ND | UNK | 0 | 3 | 0 | 3 | 0 | 3 | 0 | 3 | 0 | 1 | 0 | 1 |
| | | | 01228 | UNK | 0 | 0 | 0 | 0 | 0 | 1 | 0 | 1 | 0 | 1 | 0 | 1 |
| | | | 01229 | UNK | 0 | 0 | 0 | 0 | 0 | 1 | 0 | 1 | 0 | 1 | 0 | 1 |
| | | | 01232 | UNK | 0 | 0 | 0 | 0 | 0 | 1 | 0 | 1 | 0 | 1 | 0 | 1 |
| | | | 01233 | UNK | 0 | 0 | 0 | 0 | 0 | 1 | 0 | 1 | 0 | 1 | 0 | 1 |
| | | | 01236 | UNK | 0 | 0 | 0 | 0 | 0 | 1 | 0 | 1 | 0 | 1 | 0 | 1 |
| | | | 0125 | UNK | 0 | 1 | 0 | 1 | 0 | 1 | 0 | 1 | 1 | 0 | 0 | 0 |
| | | | 0128/20 | UNK | 0 | 0 | 0 | 0 | 0 | 1 | 0 | 1 | 0 | 1 | 0 | 1 |
| | | | SMV 4 | UNK | 0 | 0 | 0 | 0 | 0 | 1 | 0 | 1 | 0 | 1 | 0 | 1 |
| | | | SMV 5 | UNK | 0 | 0 | 0 | 0 | 0 | 1 | 0 | 1 | 0 | 1 | 0 | 1 |
| | | | SMV 6 | UNK | 0 | 0 | 0 | 0 | 0 | 1 | 0 | 1 | 0 | 1 | 0 | 1 |
| | | B/5 | 01214 | UNK | 0 | 4 | 0 | 4 | 0 | 5 | 0 | 5 | 0 | 3 | 0 | 1 |
| | | | 01231 | UNK | 0 | 0 | 0 | 0 | 0 | 1 | 0 | 1 | 0 | 1 | 0 | 1 |
| | | B/6 | 0123 | Race 1 | 0 | 1 | 0 | 1 | 0 | 2 | 0 | 2 | 0 | 2 | 0 | 2 |

*(Continued)*

| Isolates tested | | | | | Result of different Foc-TR4 diagnostic systems | | | | | | | | | | | |
|---|---|---|---|---|---|---|---|---|---|---|---|---|---|---|---|---|
| | | | | | Existing systems | | | | | | This study | | | | | |
| Status | Species[a] | Clade / Foc-SC[b] | VCG[a] | Race[a] | qPCR1[c] | | qPCR2[c] | | LAMP3[c] | | LAMP Cand-23 | | LAMP Cand-5 | | LAMP Cand-27 | |
| | | | | | +[d] | -[d] | +[d] | -[d] | +[d] | -[d] | +[d] | -[d] | +[d] | -[d] | +[d] | -[d] |
| | | B/7 | 01218 | Race 1 | 0 | 0 | 0 | 0 | 0 | 1 | 0 | 1 | 0 | 1 | 0 | 1 |
| | | | 01234 | UNK | 0 | 0 | 0 | 0 | 0 | 1 | 0 | 1 | 0 | 1 | 0 | 1 |
| | | | 01235 | UNK | 0 | 0 | 0 | 0 | 0 | 1 | 0 | 1 | 0 | 1 | 0 | 1 |
| | | | SMV 10 | UNK | 0 | 0 | 0 | 0 | 0 | 1 | 0 | 1 | 0 | 1 | 0 | 1 |
| | | B/8 | 01230 | Race 1 | 0 | 0 | 0 | 0 | 0 | 1 | 0 | 1 | 0 | 1 | 0 | 1 |
| | | C/3 | 01226 | Race 1 | 0 | 3 | 0 | 3 | 0 | 3 | 0 | 3 | 0 | 3 | 0 | 3 |
| | | UNK | | Race 1 | 0 | 5 | 0 | 5 | 0 | 5 | 0 | 5 | 0 | 2 | 0 | 0 |
| | F. fujikuroi | | | | 0 | 2 | 0 | 2 | 0 | 2 | 0 | 2 | 0 | 2 | 0 | 0 |
| | F. graminearum | | | | 0 | 1 | 0 | 1 | 0 | 1 | 0 | 1 | 1 | 0 | 0 | 0 |
| | F. oxysporum | | | Endophyte | 0 | 2 | 0 | 2 | 1 | 2 | 0 | 3 | 0 | 3 | 0 | 2 |
| | | | | UNK | 0 | 7 | 0 | 7 | 0 | 8 | 0 | 8 | 0 | 7 | 0 | 1 |
| | F. oxysporum f. sp. cepae | | | | 0 | 2 | 0 | 2 | 0 | 2 | 0 | 2 | 0 | 2 | 0 | 0 |
| | F. oxysporum f. sp. cyclamini | | | | 0 | 1 | 0 | 1 | 0 | 1 | 0 | 1 | 0 | 1 | 0 | 0 |
| | F. oxysporum f. sp. lini | | | | 0 | 1 | 0 | 1 | 0 | 1 | 0 | 1 | 0 | 1 | 0 | 0 |
| | F. oxysporum f. sp. lycopersici | | | | 0 | 1 | 0 | 1 | 0 | 1 | 0 | 1 | 0 | 1 | 0 | 0 |
| | F. oxysporum f. sp. medicaginis | | | | 0 | 1 | 0 | 1 | 0 | 1 | 0 | 1 | 0 | 1 | 0 | 0 |
| | F. oxysporum f. sp. melonis | | | | 0 | 1 | 0 | 1 | 0 | 1 | 0 | 1 | 0 | 1 | 0 | 0 |
| | F. oxysporum f. sp. radicis-lycopersici | | | | 0 | 1 | 0 | 1 | 0 | 1 | 0 | 1 | 0 | 1 | 0 | 0 |
| | F. oxysporum f. sp. radicisvanilla | | | | 0 | 1 | 0 | 1 | 0 | 1 | 0 | 1 | 0 | 1 | 0 | 0 |
| | F. oxysporum f. sp. vanilla | | | | 0 | 2 | 0 | 2 | 1 | 1 | 0 | 2 | 0 | 2 | 0 | 0 |
| | F. oxysporum f. sp. vasinfectum | | | | 0 | 1 | 0 | 1 | 0 | 1 | 0 | 1 | 0 | 1 | 0 | 0 |
| | F. oxysporum Fo47 | | | | 0 | 1 | 0 | 1 | 0 | 1 | 0 | 1 | 0 | 1 | 0 | 0 |
| | F. oxysporum PHRC474 | | | | 0 | 1 | 0 | 1 | 0 | 1 | 0 | 1 | 0 | 1 | 0 | 0 |
| | Fusarium f. sp. | | | | 0 | 2 | 0 | 2 | 0 | 2 | 0 | 2 | 0 | 0 | 0 | 0 |
| | Pseudocercospora fijiensis | | | | 0 | 3 | 0 | 3 | 0 | 3 | 0 | 3 | 0 | 3 | 0 | 0 |
| | Ralstonia solanacearum | | | | 0 | 0 | 0 | 0 | 0 | 0 | 0 | 1 | 0 | 0 | 0 | 0 |
| Total Counts | | | | | 40 | 64 | 30 | 72 | 48 | 110 | 46 | 114 | 21 | 100 | 11 | 51 |

[a] Abbreviations: "F": *Fusarium*, "TR4*": TR4 verified by PCR, "N.D." Not determined, "UNK": Unknown, "SMV": Single-Member VCG.

[b] The Clade and VCG classification is based on Mostert et al., 2022 [27].

[c] qPCR1: Aguayo et al., 2017 [20]; qPCR2: Matthews et al., 2020 [21]; LAMP3: Ordoñez et al., 2019 [37].

[d] Results column meanings: +: Positive, -: Negative.

Additionally, DNA extractions of conidial dilutions mixed with healthy plant material were tested to compare the Cand-23 LAMP assay with the Ordoñez LAMP assay [37], and two qPCR assays [20,21]. For both LAMP systems tested, a LOD of $10^5$ cells. $\mu L^{-1}$ was reached. An average TTR at 16:35 min and annealing peak at 82.7 °C were recorded for Cand-23 LAMP system, while for the Ordoñez LAMP assay [37] were recorded an average TTR 15:20 min, and annealing peak at 84.5°C (S3 Fig). Collectively, these results indicated that both LAMP systems display similar sensitivity performances. Considering the qPCR assays, the Matthews et al., 2020 [21] SYBR green qPCR could not detect any of the samples

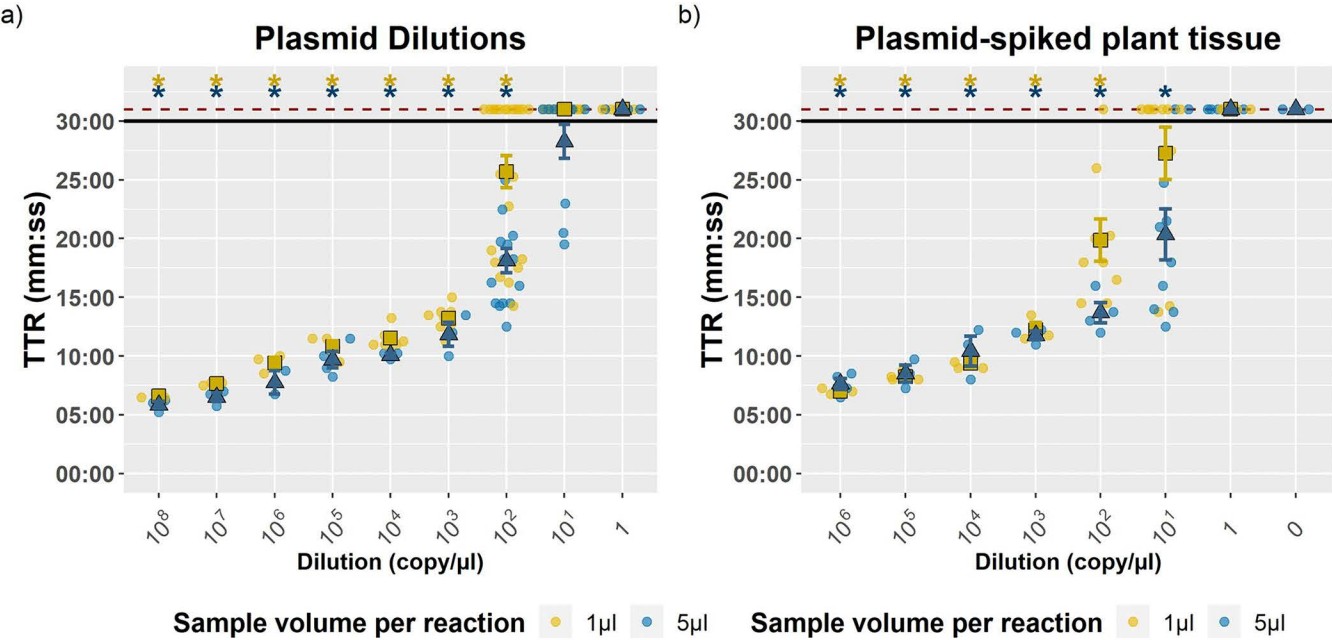

**Fig 4. Sensitivity of the Cand-23-LAMP system as assessed in plasmid dilutions (a) and in plasmid-spiked plant tissue extractions (b).** TTR: Time To Result (mins:ss). The colour and shape of the data points indicates the DNA volume used in the reaction; blue triangles (5 μL) or yellow squares (1 μL) for TTR means, while replicates are shown as solid circles. Error bars indicate the Standard Error (SE) of the mean. The horizontal black line indicates the threshold above which a sample is considered negative, and the dashed red horizontal line indicates the value assigned to the samples that produced null signal (negative). The asterisks on top indicate whether the mean is positive with a 95% of confidence level.

from this extraction, whilst the Aguayo system [20] reached a LOD of $10^5$ cells. $μL^{-1}$. The detailed results of all sensitivity tests can be found in S6 Table.

### Detection from artificially and naturally infected plant material

In glasshouse conditions, all the healthy and asymptomatic plants tested negative. In the field, the situation was more complex because the presence of other pathogens, particularly the bacterium causing Banana Blood Disease, produced confusing symptoms (yellowing and wilting of leaves, internal vascular discoloration), making the correlation between symptoms and the presence of Foc-TR4 unclear. In Java, Foc-TR4 was detected by both LAMP assays and the qPCR assay in only 10 samples out of 66, which had been collected from seven banana plants of the cultivars Pisang Barangan, Balbisiana, and Pisang Raja, located in Cianjur. A perfect correspondence of results was obtained between the Cand-23 LAMP, and the qPCR results performed on samples from Comoros and Vietnam (Fig 5). Moreover, these results agreed with the VCG determined from the isolated fungi, when available (S7 Table).

The Cohen's kappa index, reflecting the agreement between Cand-23 and qPCR, varied between different datasets (Fig 5a, Table 5 and S7 Table) from 0.65 (substantial) in the glasshouse (n = 34), 0.49 (moderate) for in-field sampling done in Indonesia (n = 66), up to 1 (perfect) in Comoros (n = 8) and Vietnam (n = 8). Considering all the field data, the kappa index was 0.64, with a total agreement of 84.1%, indicating a substantial agreement with the reference method. However, the results obtained from field data between Cand-23 LAMP and qPCR were significantly different (McNemar's test p = 0.0265), probably due to the higher positives detected by qPCR compared to LAMP. This discrepancy is reflected in the diagnostics Sensitivity (63.3%) whereas the Specificity shows consistently high values (96.4%). For the glasshouse experiment, results were similar with a total agreement of 82.4%, 0.65 kappa index, although with a non- significant

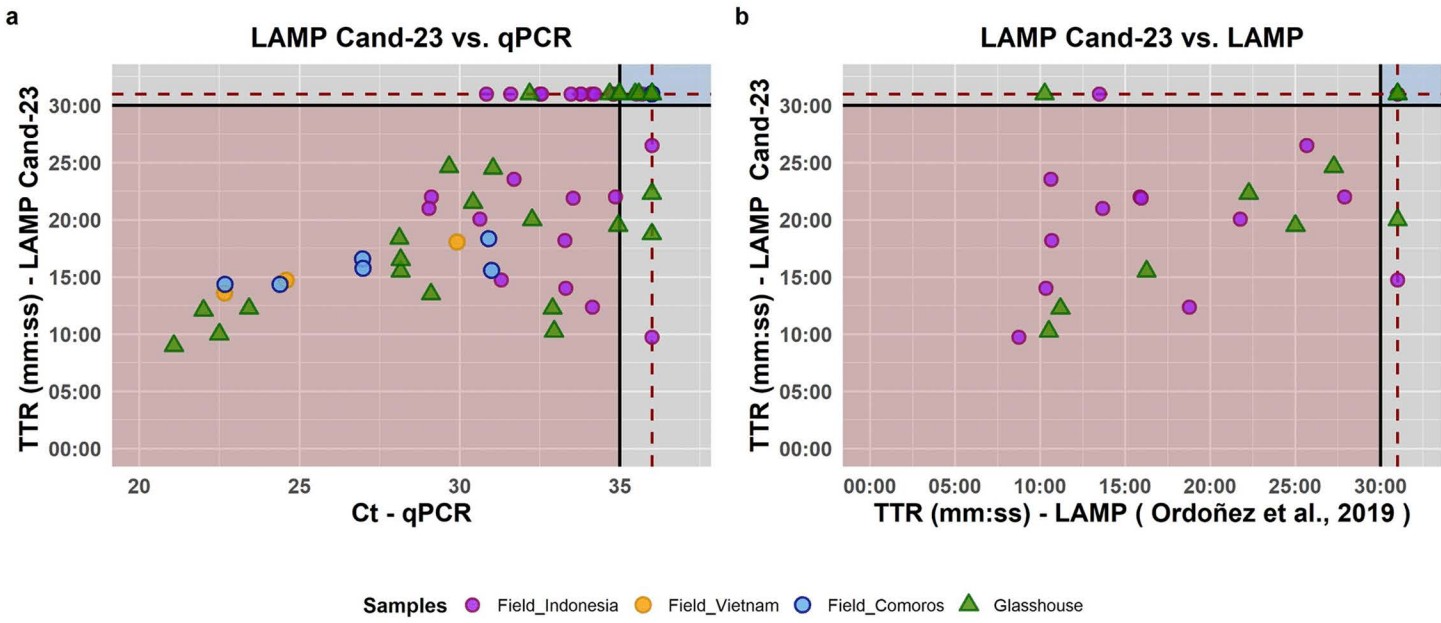

**Fig 5. Cand-23 *in planta* result comparison.** Comparison of the results obtained with the Cand-23 LAMP assay compared to the results with: (a) the Foc-TR4 specific qPCR as reported by Matthews *et al*., (2020) [21] and **(b)** Foc-TR4 specific Ordóñez LAMP assay [37]. The colour of the data points refers the origin of the data (naturally infected samples in Indonesia, Vietnam of Comoros, or artificially inoculated material in the Glasshouse) and indicates the average of the different replicates.

McNemar p-value (0.6831). The glasshouse and Indonesia samples were also tested with Ordóñez LAMP [37] to compare with Cand-23 LAMP (S7 Table and Fig 5b) No significant differences were obtained between the two methods and the different parameters reflected a high agreement between the two methods: an overall kappa coefficient of 0.87, and sensitivity and specificity values of 89.5% and 97.1%, respectively.

## Discussion

Foc-TR4 continues to expand globally, with latest detections reported in Latin America (Colombia, Peru and Venezuela) [14] and Africa (Mayotte [53], and Comoros islands [54]). The continued spread poses a significant threat to the global banana industry due to the lack of effective curative measures against the disease once established. Consequently, there is an urgent need for a rapid and reliable diagnostic tool for early detection to implement timely containment strategies and mitigate the impact of the disease. In this study, we present a new real-time LAMP assay for Foc-TR4 VCG 01213/16 identification, which was robustly designed through comparative genomic approach and specificity testing on a diverse collection of 148 Foc-TR4 genomes, and 146 non-target genomes, including all known Foc VCGs and other closely *F. oxysporum* as well as other genera strains. This assay can be applied directly in-field, and therefore offer an improved tool that can be used for early detection in surveillance efforts or in containment strategies on-farm.

In this study the K-mer Elimination by Cross-reference (KEC) method [38] was considered for robust identification of a Foc-TR4 specific sequence region. The KEC method has previously been used to develop LAMP assays in plant pathogenic bacteria and fungi [55,56], but not yet on *Fusarium oxysporum*. The KEC method enables unbiased, genome-wide identification of TR4-specific sequences by operating directly at the k-mer level, independently of gene annotation or pre-defined loci, thereby reducing reference bias compared with gene- or reference-based approaches. In addition, KEC allows the direct selection of candidate regions of a predefined minimum length, which is essential for LAMP primer design,

**Table 5. Summary of the results obtained by the Cand-23 LAMP compared to qPCR assay as published by Matthews *et al*. (2020) [21].**

| | Field_Comoros | | Field_Indonesia | | Field_Vietnam | | Glasshouse | | TOTAL FIELD | | TOTAL | |
|---|---|---|---|---|---|---|---|---|---|---|---|---|
| | LAMP+ | LAMP- | LAMP+ | LAMP- | LAMP+ | LAMP- | LAMP+ | LAMP- | LAMP+ | LAMP- | LAMP+ | LAMP- |
| qPCR + | 6 | 0 | 10 | 11 | 3 | 0 | 15 | 4 | 19 | 11 | 34 | 15 |
| qPCR - | 0 | 2 | 2 | 43 | 0 | 5 | 2 | 13 | 2 | 50 | 4 | 63 |
| n. Samples | 6 | 2 | 12 | 54 | 3 | 5 | 17 | 17 | 21 | 61 | 38 | 78 |
| Total n. Samples | 8 | | 66 | | 8 | | 34 | | 82 | | 116 | |
| Pos. agreement | 100% | | 83.3% | | 100% | | 88.2% | | 90.5% | | 89.5% | |
| Neg. agreement | 100% | | 79.6% | | 100% | | 76.5% | | 82% | | 80.8% | |
| Total agreement | 100% | | 80.3% | | 100% | | 82.4% | | 84.1% | | 83.6% | |
| **Cohen's kappa (± SD)** | **1 (±0)** | | **0.49 (±0.12)** | | **1 (±0)** | | **0.65 (±0.13)** | | **0.64 (±0.09)** | | **0.65 (±0.07)** | |
| McNemar's Chi-sq. | | | 4.92 | | | | 0.17 | | 4.92 | | 5.26 | |
| McNemar's p-value (Chi-sq.) | | | 0.0265 | | | | 0.6831 | | 0.0265 | | 0.0218 | |
| Sensitivity (SE) | 100% | | 47.6% | | 100% | | 78.9% | | 63.3% | | 69.4% | |
| Specificity (SP) | 100% | | 95.6% | | 100% | | 86.7% | | 96.2% | | 94% | |
| Accuracy (AC) | 100% | | 80.3% | | 100% | | 82.4% | | 84.1% | | 83.6% | |
| Positive Predictive Value (PPV) | 100% | | 83.3% | | 100% | | 88.2% | | 90.5% | | 89.5% | |
| Negative Predictive Value (NPV) | 100% | | 79.6% | | 100% | | 76.5% | | 82% | | 80.8% | |

Results are presented for artificially inoculation conducted under glasshouse conditions, as well as naturally infected samples analysed collected in Indonesia, Comoros, and Vietnam. Parameter description: Sensitivity (SE) = TP/ (TP+FN): proportion of true positives correctly detected by LAMP. Specificity (SP) = TN/ (TN+FP): proportion of true negatives correctly identified. Positive Predictive Value (PPV) = TP/ (TP+FP): probability that a positive LAMP result is a true positive. Negative Predictive Value (NPV) = TN/ (TN+FN): probability that a negative LAMP result is a true negative. Accuracy (AC) = (TP+TN)/ (TP+TN+FP+FN): overall proportion of correctly classified samples. Cohen's Kappa ≤ 0 corresponds to no agreement, 0.01–0.20 is "none to slight", 0.21–0.40 is "fair", 0.41–0.60 is "moderate", 0.61–0.80 is "substantial", and 0.81–1.00 is "near-perfect". A McNemar's p value below 0.05 was taken as evidence to reject the null hypothesis of equal marginal proportions, indicating a statistically significant difference in detection outcomes between the two methods compared.

where recommended target sizes range between 120 and 180 nucleotides [33]. This contrasts with other approaches used so far to develop Foc-TR4 diagnostic assays, such as DArTseq sequencing which implies genome complexity reduction [21,37], or the identification of polymorphisms in known *SIX* genes [50,57], which are biased towards enzyme restrictions sites or genes, and yield short candidate sequences. These sequences often require to be extended to allow the design of LAMP primers, at the risk of losing the specificity. Several alternative marker discovery strategies were evaluated in this study, including ortho-group filtering, and Illumina read mapping and filtering against a TR4 reference genome. But these approaches yielded few candidates and lacked sufficient specificity or inclusivity. In contrast, the KEC method generated multiple candidate regions, enabling downstream filtering and experimental validation to select a robust marker. Notwithstanding this advantage, the KEC method may also be limited by the quality of the input genome assemblies used, which might include poor genomic sequences such as high GC content regions, not appropriate for primer design. To overcome these limitations the reference genome (referred as "master" genome in KEC protocol) used was a high quality long-read assembly of an Foc-TR4 isolate. The repetitive element-masked version of the genome was used to avoid the selection of candidates within the repeat-rich regions of the genome which could potentially be unstable. One disadvantage of the KEC approach employed however, was that not all the predicted sequences were target specific, as BLAST searches showed matches in non-target genomes. The initial version (v1.0) of the KEC software was used in this analysis, in which the *include* mode applied less stringent criteria for sequence uniqueness than in later releases, potentially allowing retention of sequences not exclusively specific to the target. This limitation was nevertheless addressed by the addition of internal and external BLAST steps improved specificity, but full inclusivity on all the target Foc-TR4 genomes was still not achieved.

This was probably due to the incompleteness of some Illumina assemblies, as regions with low read depth and quality were excluded during the assembly process [58]. Adding the Illumina read mapping step allowed to identify genuine genomic absences on certain Foc-TR4 isolates in some of the candidates, leading to the selection of the Candidate 23, which was present in all 120 target genomes and absent in all non-target genomes representing all known *Foc* VCGs, as well as non-pathogenic and endophytic *F. oxysporum* species associated with banana. This multi-step validation demonstrated that the KEC-based strategy, combined with downstream filtering, provided a clear improvement in marker specificity and inclusivity compared to alternative approaches cited above. This was demonstrated experimentally when tested on a large specificity panel comprising Foc-TR4 isolates representatives from all known other *Foc* VCGs, additional isolates belonging to the *F. oxysporum* species complex and fungal endophytes isolates from banana environment.

The selected Candidate 23 is located within a unique intergenic region on Chromosome 2 of the reference II5 (GCA 031834405). The sequence was characterized by the presence of repetitive nucleotides. Additionally, read coverage was low, with a pronounced decrease observed in one part of the sequence. Due to the presence of repetitive nucleotides in the Candidate 23 target sequence, the design of LAMP primers with primer design software was not possible. This issue was resolved through the manual design of a loop primer and Swarm primers, a lesser-known alternative to loop primers that enhanced the LAMP reaction [52,59]. This allowed the development of a LAMP system that worked and provided a clear single annealing peak.

This study also analysed the specificity of other already published Foc-TR4 LAMP assays using the *in-silico* approach. Amplicon sequences from Li *et al.* (2013) [19] showed good inclusivity in terms of BLAST and alignment on all Foc-TR4 Illumina genomes, but also had high (95–100%) identity and full alignment with 43 and 25 non-target genomes respectively, including 100% BLAST identity on three VCG 0121 genomes. A similar result was obtained with the LAMP assay designed by Canare *et al.* (2024) [50], consistent with the fact that this LAMP targets the *SIX8a* homologue gene, which is present in all Race 4 isolates [28,60]. In contrast, the Ordóñez *et al.* (2019) LAMP assay [37] was not fully inclusive, not picking up two Foc-TR4 genomes (both of Chinese origin). Whether this was a genuine lack or an artifact due to sequencing issues on these genomes, could not be validated on DNA; however, the correlation between the *in silico* and *in vitro* was good in the sixty-two isolates for which there was data and DNA. Also, similar cases of missing alignments on genomes for Candidate 5 were validated experimentally on seven DNA isolates. On the other hand, this LAMP system was not fully exclusive, showing high sequence homology to an *F. oxysporum* endophyte, and positives on non-target DNA isolates. The latter result stresses the importance of performing thorough specificity testing especially considering endophytic, non-pathogenic isolates which might amplify as false positives. This is particularly challenging in fungal species with a broad host range such as *F. oxysporum* given the genetic similarity between non-pathogenic and endophytic isolates with pathogenic isolates [61].

The sensitivity of our LAMP assay obtained a consistent LOD of up to 100 copies per reaction with plasmid dilutions, which is the same dilution as reported in other existing LAMP systems for Foc-TR4 [40]. Interestingly a better LOD of 50 copies per reaction was reached when testing the plasmid dilutions in a plant tissue background extraction. Plant tissue providing a matrix which would help plasmid DNA precipitation and binding, preventing degradation and DNA loss could possibly explain this difference with the tests carried out on simple water plasmid dilutions [62,63]. Nevertheless, this corroborates the fact that non-target DNA presence does not affect the assay performance.

The LAMP assay designed in this study was evaluated under glasshouse and field conditions across three countries (Indonesia, Vietnam, and Comoros), and it performed reliably using both commercial and simplified DNA extraction protocols. The tool's portability was also demonstrated through successful implementation in non-laboratory settings, highlighting its potential for use in resource-limited settings.

In comparison with the qPCR assays tested in parallel, our LAMP assay showed strong correlation; however, qPCR still proved to remain as the gold standard in terms of sensitivity. Nevertheless, this came with the trade-off of higher requirements in terms of laboratory, extraction equipment and time.

## Conclusion

Extensive regulatory measures have been implemented to prevent the introduction and establishment of Foc-TR4 in unaffected areas. The effectiveness of this surveillance relies heavily on having specific and sensitive detection protocols. Although molecular-based methods are available, many have demonstrated issues with specificity. A novel highly specific LAMP assay was developed in this study for in-field detection of Foc-TR4, which was validated by comparative genomics approach. This approach accounts for the known genetic diversity within Foc and FOSC, improving upon the specificity of previously published Foc-TR4 detection assays to date. The LAMP assay was shown to be highly effective in detecting FocTR4 in infected banana plants, particularly in non-laboratory environments, offering a promising tool for improving disease management strategies. Implementing this technique regularly in diagnostic processes will allow early identification of infected banana plants, facilitating timely intervention to prevent widespread outbreaks. The Cand-23 LAMP assay is now commercialized in a ready-to-use "LAMP Foc-TR4 kit" produced by QUALIPLANTE SAS (Lavérune, France. https://www.qualiplante.eu), in an all-inclusive format containing the lyophilized primer mix, the lyophilized enzyme, a resuspension buffer, and positive and negative controls.

## Supporting information

**S1 Methods. DNA extraction protocol.**
(DOCX)

**S1 Fig. Plasmid diagram.**
(TIF)

**S2 Fig. Primer optimisation recipe comparison test.** The red triangle shows the mean of each recipe. Cand23: original concentration. 2x_Cand23: double concentration of the original.
(TIF)

**S3 Fig. Sensitivity results on conidia dilutions-spiked plant tissue. a: with LAMP systems. b: using qPCR systems.** TTR: Time To Result (mins:ss). The colour of the data points indicates the DNA volume used in the reaction, means are indicated by blue triangle (5µl) or yellow square (1µl), while replicates are solid circles. Error bars indicate the Standard Error (SE) of the mean. The horizontal black line indicates the threshold above which a sample is considered negative, and the dashed red horizontal line indicates the value assigned to the samples that produced null signal (negative). The asterisks on top indicate for each DNA volume group, whether the mean is positive with a 95% of confidence level.
(TIF)

**S1 Table. Detailed information of the genomic data used in this study.**
(XLSX)

**S2 Table. Detailed parameters of the KEC candidates and their derived primers.**
(XLSX)

**S3 Table. Detailed information of the DNA isolates used in this study.**
(XLSX)

**S4 Table. Detailed informations on the LAMP candidates as compared with LAMP and PCR/qPCR assays already published.**
(XLSX)

**S5 Table. Detailed Specificity Results of our LAMP candidates compared with qPCRs and LAMP references.**
(XLSX)

**S6 Table. Detailed Sensitivity Results of our LAMP candidates compared with qPCRs and LAMP references.**
(XLSX)

**S7 Table. Detailed TR4 Diagnostic results of test done on plants of our LAMP candidates compared with qPCR and LAMP references.**
(XLSX)

## Acknowledgments

We thank Natalya Saveljeva, Veronique Roussel and Christine Pages for technical assistance, Lorène Milliex for the project management, Caroline Chatillon for assistance in Qualiplante, Megan Matthews for help in extractions and assistance. We also thank Nguyen Thi Thô for assistance and extractions, and Mauricio Soto-Suarez (Agrosavia) for having shared Foc-TR4 genome assemblies and long-reads ONT data from Colombia.

## Author contributions

**Conceptualization:** Mikel Arrieta Salgado, Camilo Gianinazzi, Emmanuel Wicker, Isabelle Robène.

**Data curation:** Mikel Arrieta Salgado, Sebastien Ravel, Samuel Rozsasi, Henri Adreit, Beatrix Coetzee, Emmanuel Wicker, Isabelle Robène.

**Formal analysis:** Mikel Arrieta Salgado, Sandrine Fabre, J. Jansen van Vuuren, Sheryl Le Roux, Isabelle Robène.

**Funding acquisition:** Altus Viljoen, Camilo Gianinazzi, Emmanuel Wicker, Isabelle Robène.

**Investigation:** Mikel Arrieta Salgado, Diane Mostert, Veronique Maillot-Lebon, Catur Hermanto, Nadia Adjanoh-Lubin, Babitha Fenelon, Yolande Chilin-Charles, Emmanuel Wicker, Isabelle Robène.

**Methodology:** Mikel Arrieta Salgado, Diane Mostert, Sebastien Ravel, Samuel Rozsasi, Veronique Maillot-Lebon, Babitha Fenelon, Yolande Chilin-Charles, Camilo Gianinazzi, Emmanuel Wicker, Isabelle Robène.

**Project administration:** Camilo Gianinazzi, Emmanuel Wicker, Isabelle Robène.

**Resources:** Diane Mostert, Agus Sutanto, Catur Hermanto, Mouzdalifa Mmadi, Abdou Azali Hamza, Nadia Adjanoh-Lubin, Henri Adreit, Sandrine Fabre, Beatrix Coetzee, Altus Viljoen, Camilo Gianinazzi, Emmanuel Wicker, Isabelle Robène.

**Software:** Sebastien Ravel.

**Supervision:** Diane Mostert, Jean Carlier, Camilo Gianinazzi, Emmanuel Wicker, Isabelle Robène.

**Validation:** Mikel Arrieta Salgado, Samuel Rozsasi, Veronique Maillot-Lebon, Agus Sutanto, Altus Viljoen, Yolande Chilin-Charles, Emmanuel Wicker, Isabelle Robène.

**Visualization:** Mikel Arrieta Salgado, Samuel Rozsasi, Emmanuel Wicker, Isabelle Robène.

**Writing – original draft:** Mikel Arrieta Salgado, Emmanuel Wicker, Isabelle Robène.

**Writing – review & editing:** Mikel Arrieta Salgado, Diane Mostert, Beatrix Coetzee, Altus Viljoen, Emmanuel Wicker, Isabelle Robène.

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
