## [Decision Letter · Decision Letter 0]

15 Jan 2026

PONE-D-25-47570Comparative genomics-based development of a LAMP assay for rapid and reliable in-field detection of Fusarium oxysporum f. sp. cubense tropical race 4PLOS One

Dear Dr. Wicker,

Thank you for submitting your manuscript to PLOS ONE. After careful consideration, we feel that it has merit but does not fully meet PLOS ONE’s publication criteria as it currently stands. Therefore, we invite you to submit a revised version of the manuscript that addresses the points raised during the review process.

We look forward to receiving your revised manuscript.

Kind regards,

Ravinder Kumar, Ph.D.

Academic Editor

PLOS One

Journal Requirements:

“This study was funded by the Projet France-Relance (PFR) CIRAD-QUALIPLANTE (2022-2024), and by the European Union Horizon 2020 Research and Innovation Program Marie Sklodowska Curie Fellowship, through the INDICANTS project grant nr 890856.”

“This study was funded by the Projet France-Relance (PFR) CIRAD-QUALIPLANTE (2022-2024), and by the European Union Horizon 2020 Research and Innovation Program Marie Sklodowska-Curie Fellowship, through the INDICANTS project grant nr 890856.”

“This study was funded by the Projet France-Relance (PFR) CIRAD-QUALIPLANTE (2022-2024), and by the European Union Horizon 2020 Research and Innovation Program Marie Sklodowska-Curie Fellowship, through the INDICANTS project grant nr 890856.”

“This study was funded by the Projet France-Relance (PFR) CIRAD-QUALIPLANTE (2022-2024), and by the European Union Horizon 2020 Research and Innovation Program Marie Sklodowska-Curie Fellowship, through the INDICANTS project grant nr 890856.”

We note that one or more of the authors is affiliated with the funding organization, indicating the funder may have had some role in the design, data collection, analysis or preparation of your manuscript for publication; in other words, the funder played an indirect role through the participation of the co-authors. If the funding organization did not play a role in the study design, data collection and analysis, decision to publish, or preparation of the manuscript and only provided financial support in the form of authors' salaries and/or research materials, please do the following:

1. Review your statements relating to the author contributions, and ensure you have specifically and accurately indicated the role(s) that these authors had in your study. These amendments should be made in the online form.

2. Confirm in your cover letter that you agree with the following statement, and we will change the online submission form on your behalf:

“The funder provided support in the form of salaries for authors [insert relevant initials], but did not have any additional role in the study design, data collection and analysis, decision to publish, or preparation of the manuscript. The specific roles of these authors are articulated in the ‘author contributions’ section.

8. Please ensure that you refer to Figure 3 in your text as, if accepted, production will need this reference to link the reader to the figure.

9. We note you have included a table to which you do not refer in the text of your manuscript. Please ensure that you refer to Table 4 in your text; if accepted, production will need this reference to link the reader to the Table.

10. Thank you for providing your underlying data as Supporting Information.

We note that the data set contains text or data that is not in English. Please note that PLOS is an English-language publisher, so we require data sets to be provided in English as well. Please upload an English-language version of your data set.

This will also allow us to determine if your data follows PLOS standards per our Data Availability policy here: https://journals.plos.org/plosone/s/data-availability

Reviewers' comments:

Reviewer's Responses to Questions

**Comments to the Author**

1. Is the manuscript technically sound, and do the data support the conclusions?

Reviewer #1: Yes

Reviewer #2: Yes

2. Has the statistical analysis been performed appropriately and rigorously? 

Reviewer #1: Yes

Reviewer #2: Yes

3. Have the authors made all data underlying the findings in their manuscript fully available?

The PLOS Data policy requires authors to make all data underlying the findings described in their manuscript fully available without restriction, with rare exception (please refer to the Data Availability Statement in the manuscript PDF file). The data should be provided as part of the manuscript or its supporting information, or deposited to a public repository. For example, in addition to summary statistics, the data points behind means, medians and variance measures should be available. If there are restrictions on publicly sharing data—e.g. participant privacy or use of data from a third party—those must be specified.requires authors to make all data underlying the findings described in their manuscript fully available without restriction, with rare exception (please refer to the Data Availability Statement in the manuscript PDF file). The data should be provided as part of the manuscript or its supporting information, or deposited to a public repository. For example, in addition to summary statistics, the data points behind means, medians and variance measures should be available. If there are restrictions on publicly sharing data—e.g. participant privacy or use of data from a third party—those must be specified.requires authors to make all data underlying the findings described in their manuscript fully available without restriction, with rare exception (please refer to the Data Availability Statement in the manuscript PDF file). The data should be provided as part of the manuscript or its supporting information, or deposited to a public repository. For example, in addition to summary statistics, the data points behind means, medians and variance measures should be available. If there are restrictions on publicly sharing data—e.g. participant privacy or use of data from a third party—those must be specified.requires authors to make all data underlying the findings described in their manuscript fully available without restriction, with rare exception (please refer to the Data Availability Statement in the manuscript PDF file). The data should be provided as part of the manuscript or its supporting information, or deposited to a public repository. For example, in addition to summary statistics, the data points behind means, medians and variance measures should be available. If there are restrictions on publicly sharing data—e.g. participant privacy or use of data from a third party—those must be specified.

Reviewer #1: No

Reviewer #2: Yes

4. Is the manuscript presented in an intelligible fashion and written in standard English?

Reviewer #1: Yes

Reviewer #2: Yes

5. Review Comments to the Author

Reviewer #1: MS on “Comparative genomics-based development of a LAMP assay for rapid and reliable infield detection of Fusariumoxysporum f. sp. cubense tropical race 4’ has been written well.

This reports LAMP (loop-mediated isothermal amplification) assay for quick field detection ofFusariumoxysporum f. sp. cubense Tropical Race 4 (Foc TR4), the causal agent of Fusarium wilt in bananas. The assay demonstrated high specificity and accurately identified the pathogen in both artificial and field samples. Overall, the LAMP test is a robust, reliable, and practical tool for on-site diagnosis of Foc TR4 infections in banana plants.

Table 1 and Table 4: Error in writing (f.sp.) should be corrected as f.sp.

In Table 1 and Table 4 under race column many rows are blank which need to be filled. If anyone is not known is to be given as unknown

Line 279: Define 10^0 clearly. is it possible to prepare serial dilution till to get nil conidia? Or it is mere control.

In the section “In-field deployment and detection from naturally infested plant material” (line 303), further details are required. Specifically, was internal severity of Foc assessed for all the samples? Additionally, please indicate whether the varieties grown were Cavendish or non-Cavendish, and specifically their genome.

Is this LAMP assay able to give inoculum load in soil? If yes, whether it was validated with spore count?

All the supplementary files could not be accessed which need to be given in easy accessible form.

Legends for tables and figures may be arranged at the end of the manuscript.

Reviewer #2: The manuscript presents a comprehensive and methodologically robust study on the development of a LAMP-based assay for the rapid detection of Fusarium oxysporum f. sp. cubense Tropical Race 4 (Foc-TR4). The topic is highly relevant to global banana production and plant pathology. The authors have employed a comparative genomics approach using a large genome dataset (294 genomes) to identify unique diagnostic targets, followed by extensive validation using 161 isolates and successful field deployment in Indonesia.

Overall, the study is well-designed, technically sound, and contributes valuable diagnostic tools for Foc-TR4 detection. However, the manuscript requires minor revisions to address typographical, formatting, and clarity issues prior to publication.

Specific Comments

The manuscript should emphasize how the KEC-based marker identification improves upon previous marker discovery methods and, where possible, quantify the comparative improvement in specificity and sensitivity.

Table 1 is excessively detailed for the main text; consider moving the comprehensive genome list to the Supplementary Material.

Remove the period (".") preceding units (e.g., µL⁻¹, mL) throughout the text.

Line 34: The sentence appears incomplete; please revise for clarity.

Line 54: Replace “The causing agent is Fusarium oxysporum f. sp. cubense” with “The causal agent is Fusarium oxysporum f. sp. cubense.”

Line 59: Replace “export” with “exports.”

Lines 79–81: Rephrase for clarity and readability.

Line 249: Replace “grinded” with “ground.”

Line 272: Replace “eluded” with “eluted.”

Line 298: Replace “sensibility” with “sensitivity.”

Line 359: The statement is unclear; please revise to improve clarity.

Ensure consistent use of µL⁻¹ (superscript minus) or “per µL” throughout the text.

Standardize spelling conventions (e.g., visualized vs visualised, optimized vs optimised) and apply consistently.

Italicize all scientific names (e.g., F. oxysporum, F. cubense).

Use en-dashes consistently in numerical ranges (e.g., “30–40°C” instead of “30-40°C”).

A thorough proofreading is recommended to correct typographical and grammatical errors throughout the manuscript.

6. PLOS authors have the option to publish the peer review history of their article (what does this mean?). If published, this will include your full peer review and any attached files.). If published, this will include your full peer review and any attached files.). If published, this will include your full peer review and any attached files.). If published, this will include your full peer review and any attached files.

...

Reviewer #1: No

Reviewer #2: **Yes:** SELVARAJAN RAMASAMYSELVARAJAN RAMASAMYSELVARAJAN RAMASAMYSELVARAJAN RAMASAMY

---

## [Author Response · Author response to Decision Letter 1]

28 Jan 2026

Dear Academic Editor and reviewers,

The comments and points raised were all considered. We wrote a point-to-point detailed response letter to both the Journal Requirements and the Reviewers comments, available in the file "Response to Reviewers" enclosed.

In the name of all co-authors, I would like to thank you for your careful reviews which greatly contributed to improve this manuscript.

Yours sincerely,

E. Wicker

---

## [Decision Letter · Decision Letter 1]

25 Feb 2026

PONE-D-25-47570R1Comparative genomics-based development of a LAMP assay for rapid and reliable in-field detection of Fusarium oxysporum f. sp. cubense tropical race 4PLOS One

Dear Dr. Wicker,

Thank you for submitting your manuscript to PLOS ONE. After careful consideration, we feel that it has merit but does not fully meet PLOS ONE’s publication criteria as it currently stands. Therefore, we invite you to submit a revised version of the manuscript that addresses the points raised during the review process.

We look forward to receiving your revised manuscript.

Kind regards,

Ravinder Kumar, Ph.D.

Academic Editor

PLOS One

Journal Requirements:

Reviewers' comments:

Reviewer's Responses to Questions

**Comments to the Author**

1. If the authors have adequately addressed your comments raised in a previous round of review and you feel that this manuscript is now acceptable for publication, you may indicate that here to bypass the “Comments to the Author” section, enter your conflict of interest statement in the “Confidential to Editor” section, and submit your "Accept" recommendation.

Reviewer #1: (No Response)

2. Is the manuscript technically sound, and do the data support the conclusions?

Reviewer #1: Yes

3. Has the statistical analysis been performed appropriately and rigorously? 

Reviewer #1: Yes

4. Have the authors made all data underlying the findings in their manuscript fully available?

The PLOS Data policy requires authors to make all data underlying the findings described in their manuscript fully available without restriction, with rare exception (please refer to the Data Availability Statement in the manuscript PDF file). The data should be provided as part of the manuscript or its supporting information, or deposited to a public repository. For example, in addition to summary statistics, the data points behind means, medians and variance measures should be available. If there are restrictions on publicly sharing data—e.g. participant privacy or use of data from a third party—those must be specified.requires authors to make all data underlying the findings described in their manuscript fully available without restriction, with rare exception (please refer to the Data Availability Statement in the manuscript PDF file). The data should be provided as part of the manuscript or its supporting information, or deposited to a public repository. For example, in addition to summary statistics, the data points behind means, medians and variance measures should be available. If there are restrictions on publicly sharing data—e.g. participant privacy or use of data from a third party—those must be specified.requires authors to make all data underlying the findings described in their manuscript fully available without restriction, with rare exception (please refer to the Data Availability Statement in the manuscript PDF file). The data should be provided as part of the manuscript or its supporting information, or deposited to a public repository. For example, in addition to summary statistics, the data points behind means, medians and variance measures should be available. If there are restrictions on publicly sharing data—e.g. participant privacy or use of data from a third party—those must be specified.requires authors to make all data underlying the findings described in their manuscript fully available without restriction, with rare exception (please refer to the Data Availability Statement in the manuscript PDF file). The data should be provided as part of the manuscript or its supporting information, or deposited to a public repository. For example, in addition to summary statistics, the data points behind means, medians and variance measures should be available. If there are restrictions on publicly sharing data—e.g. participant privacy or use of data from a third party—those must be specified.

Reviewer #1: Yes

5. Is the manuscript presented in an intelligible fashion and written in standard English?

Reviewer #1: Yes

6. Review Comments to the Author

Reviewer #1: MS "Comparative genomics-based development of a LAMP assay for rapid and reliable in-field detection of Fusarium oxysporum f. sp. cubense tropical race 4" has been improvised. However the minor annotations (italics, typo errors etc) as noted in the MS need to be taken care.

7. PLOS authors have the option to publish the peer review history of their article (what does this mean?). If published, this will include your full peer review and any attached files.). If published, this will include your full peer review and any attached files.). If published, this will include your full peer review and any attached files.). If published, this will include your full peer review and any attached files.

...

Reviewer #1: No

---

## [Author Response · Author response to Decision Letter 2]

4 Mar 2026

Dear Reviewer,

Following your comments, the main text has been carefully checked, and typo errors were identified and corrected. Some sentences were aso rephrased to gain in clarity. See the "Response to Reviewers" letter for details.

---

## [Decision Letter · Decision Letter 2]

6 Apr 2026

Comparative genomics-based development of a LAMP assay for rapid and reliable in-field detection of Fusarium oxysporum f. sp. cubense tropical race 4

PONE-D-25-47570R2

Dear Dr. Wicker,

We’re pleased to inform you that your manuscript has been judged scientifically suitable for publication and will be formally accepted for publication once it meets all outstanding technical requirements.

Kind regards,

Ravinder Kumar, Ph.D.

Academic Editor

PLOS One

**Comments to the Author**

1. If the authors have adequately addressed your comments raised in a previous round of review and you feel that this manuscript is now acceptable for publication, you may indicate that here to bypass the “Comments to the Author” section, enter your conflict of interest statement in the “Confidential to Editor” section, and submit your "Accept" recommendation.

Reviewer #1: All comments have been addressed

2. Is the manuscript technically sound, and do the data support the conclusions?

Reviewer #1: Yes

3. Has the statistical analysis been performed appropriately and rigorously? 

Reviewer #1: N/A

4. Have the authors made all data underlying the findings in their manuscript fully available?

The PLOS Data policy requires authors to make all data underlying the findings described in their manuscript fully available without restriction, with rare exception (please refer to the Data Availability Statement in the manuscript PDF file). The data should be provided as part of the manuscript or its supporting information, or deposited to a public repository. For example, in addition to summary statistics, the data points behind means, medians and variance measures should be available. If there are restrictions on publicly sharing data—e.g. participant privacy or use of data from a third party—those must be specified.requires authors to make all data underlying the findings described in their manuscript fully available without restriction, with rare exception (please refer to the Data Availability Statement in the manuscript PDF file). The data should be provided as part of the manuscript or its supporting information, or deposited to a public repository. For example, in addition to summary statistics, the data points behind means, medians and variance measures should be available. If there are restrictions on publicly sharing data—e.g. participant privacy or use of data from a third party—those must be specified.requires authors to make all data underlying the findings described in their manuscript fully available without restriction, with rare exception (please refer to the Data Availability Statement in the manuscript PDF file). The data should be provided as part of the manuscript or its supporting information, or deposited to a public repository. For example, in addition to summary statistics, the data points behind means, medians and variance measures should be available. If there are restrictions on publicly sharing data—e.g. participant privacy or use of data from a third party—those must be specified.requires authors to make all data underlying the findings described in their manuscript fully available without restriction, with rare exception (please refer to the Data Availability Statement in the manuscript PDF file). The data should be provided as part of the manuscript or its supporting information, or deposited to a public repository. For example, in addition to summary statistics, the data points behind means, medians and variance measures should be available. If there are restrictions on publicly sharing data—e.g. participant privacy or use of data from a third party—those must be specified.

Reviewer #1: Yes

5. Is the manuscript presented in an intelligible fashion and written in standard English?

Reviewer #1: Yes

6. Review Comments to the Author

Reviewer #1: The manuscript on "Comparative genomics-based development of a LAMP assay for rapid and reliable in-field detection of Fusarium oxysporum f. sp. cubense tropical race 4" is in acceptable form.

7. PLOS authors have the option to publish the peer review history of their article (what does this mean?). If published, this will include your full peer review and any attached files.). If published, this will include your full peer review and any attached files.). If published, this will include your full peer review and any attached files.). If published, this will include your full peer review and any attached files.

...

Reviewer #1: No

---

## [Editor Report · Acceptance letter]

PONE-D-25-47570R2

PLOS One

Dear Dr. Wicker,

I'm pleased to inform you that your manuscript has been deemed suitable for publication in PLOS One. Congratulations! Your manuscript is now being handed over to our production team.

Kind regards,

on behalf of

Dr. Ravinder Kumar

Academic Editor

PLOS One